# PYK2 senses calcium through a disordered dimerization and calmodulin-binding element

Afaque A. Momin [1,2], Tiago Mendes[3,5], Philippe Barthe[4,5], Camille Faure[3], SeungBeom Hong[1,2], Piao Yu[1,2], Gress Kadaré[3], Mariusz Jaremko[2], Jean-Antoine Girault [3], Łukasz Jaremko[2] & Stefan T. Arold [1,2,4✉]

Multidomain kinases use many ways to integrate and process diverse stimuli. Here, we investigated the mechanism by which the protein tyrosine kinase 2-beta (PYK2) functions as a sensor and effector of cellular calcium influx. We show that the linker between the PYK2 kinase and FAT domains (KFL) encompasses an unusual calmodulin (CaM) binding element. PYK2 KFL is disordered and engages CaM through an ensemble of transient binding events. Calcium increases the association by promoting structural changes in CaM that expose auxiliary interaction opportunities. KFL also forms fuzzy dimers, and dimerization is enhanced by CaM binding. As a monomer, however, KFL associates with the PYK2 FERM-kinase fragment. Thus, we identify a mechanism whereby calcium influx can promote PYK2 self-association, and hence kinase-activating *trans*-autophosphorylation. Collectively, our findings describe a flexible protein module that expands the paradigms for CaM binding and self-association, and their use for controlling kinase activity.

[1] Computational Bioscience Research Center (CBRC), King Abdullah University of Science and Technology (KAUST), Thuwal 23955-6900, Saudi Arabia. [2] Bioscience Program, Division of Biological and Environmental Science and Engineering (BESE), King Abdullah University of Science and Technology (KAUST), Thuwal 23955-6900, Saudi Arabia. [3] Inserm UMR-S 1270, Sorbonne Université, Faculty of Sciences and Engineering, Institut du Fer à Moulin, 75005 Paris, France. [4] Centre de Biologie Structurale (CBS), University Montpellier, INSERM U1054, CNRS UMR 5048, 34090 Montpellier, France. [5] These authors contributed equally: Tiago Mendes, Philippe Barthe. ✉email: stefan.arold@kaust.edu.sa

Cellular signal transduction relies to a large extent on the capacity of multidomain protein kinases to sense, process, and transduce specific information provided by proteins, small molecules or ions. Focal adhesion kinase (FAK) and protein tyrosine kinase 2-beta (PYK2) are two closely related non-receptor protein tyrosine kinases that play vital roles in various cellular functions, including motility, adhesion, signalling, and gene expression[1–6]. Further, they are important drivers of cancer cell invasiveness[7–9].

The PYK2-encoding *PTK2B* gene originated via the duplication and diversification of the FAK-encoding *PTK2* gene in vertebrates[10]. Both protein paralogs have the same domain organisation: an N-terminal band 4.1, ezrin, radixin, moesin (FERM) domain, a tyrosine kinase domain, and a C-terminal focal adhesion targeting (FAT) domain (Fig. 1a). These domains are linked by flexible regions containing several short linear interaction motifs and sites for post-translational modifications, particularly phosphorylation[2,5,11–16]. The three folded domains are markedly better conserved (47%–60% sequence identity) than the linker regions (20%–30% identity; Fig. 1a)[16].

This shared structural composition results in a significant similarity in the activation mechanisms and biological functions of PYK2 and FAK. Indeed, both proteins have an autoinhibited conformation in which the FERM domain binds to the kinase domain, thereby inhibiting its catalytic function[17,18]. The activation of kinase-dependent functions requires ligand-induced self-association (dimerisation or clustering) to trigger autophosphorylation in trans of a pivotal tyrosine (Y397 in FAK and Y402 in PYK2). This tyrosine is situated in the region linking the FERM and kinase domains[16,19–23]. Once phosphorylated, this tyrosine and an adjacent proline-rich motif form a bidentate-binding site for the SH2–SH3 domain fragments of Src kinases (Src and Fyn)[24]. This bidentate interaction activates Src kinases, which in turn contribute toward most of the catalytic activity associated with the catalytically active Src–FAK or Src–PYK2 complexes[20].

In line with their evolutionary history and preserved domain structure, FAK and PYK2 have overlapping cellular functions, for example in cellular adhesion[1,2,25], growth factor receptor signalling[26], kinase-independent nuclear effects on gene expression, and p53 degradation[27]. However, PYK2 is also functionally distinct from FAK, particularly based on its unique role in $Ca^{2+}$ sensing[11,28,29]. This role allows PYK2 to partake in signalling pathways triggered by elevated $Ca^{2+}$ levels in cells, such as cellular stress initiated by extracellular signals[16,30,31]. PYK2 acts as a central transducer of $Ca^{2+}$ signals at cell contacts[29] and in other loci[32]. In contrast to FAK, which is found in many cell types and is abundantly expressed during development, PYK2 is highly enriched in the adult forebrain[33]. In neurons, PYK2 is activated by $Ca^{2+}$ influx following the depolarisation or activation of glutamate receptors[34–36]. In particular, PYK2 activation by $Ca^{2+}$ promotes neurite outgrowth in neurons and is required for synaptic plasticity[30]. Moreover, PYK2 accumulates in the nucleus following $Ca^{2+}$ influx[37], a step regulated by its phosphorylation[38]. These differences in gene expression patterns, protein sequences, and $Ca^{2+}$ sensing limit the functional redundancy of FAK and PYK2, and give rise to distinct and even antagonistic functions[9,39].

Although it is generally accepted that $Ca^{2+}$ sensing by PYK2 is at least in part mediated by calmodulin (CaM), the underlying mechanism remains poorly understood and controversial. Independent investigations by two research groups led to the proposition of two different mechanisms: Kohno et al. proposed that $Ca^{2+}$-loaded CaM ($Ca^{2+}$/CaM) binds to a helix within the FERM domain, whereas Xie et al. suggested that $Ca^{2+}$/CaM binds to a helical region of the kinase domain (Fig. 1a)[40,41]. Each proposed site would require unfolding of either the FERM or the kinase domain, respectively, to be accessible for canonical CaM

interactions (Supplementary Fig. 1a)[42], but neither group provided a structural basis for their mechanism. In this study, we used a multidisciplinary approach to determine the molecular mechanism of $Ca^{2+}$ sensing by PYK2. Our results provide converging evidence regarding the architecture and role of a non-conventional $Ca^{2+}$/CaM-binding site located in a region situated between the PYK2 kinase and FAT domains.

## Results

### The direct CaM-binding element resides in the kinase-FAT linker (KFL). 
First, we probed the association between PYK2 and CaM using HEK239 cells transfected with GFP-fused PYK2 plasmid constructs (Fig. 1b). These cells did not express detectable amounts of endogenous PYK2 (Fig. 1b). Cell lysates were incubated with CaM agarose beads in the presence of $Ca^{2+}$ or EGTA and binding between GFP and PYK2 was analysed using pulldown assays followed by western blotting using an anti-GFP antibody. Although full-length PYK2 bound strongest, all constructs containing the FERM domain also associated with CaM beads significantly better than GFP alone. Additionally, the kinase–FAT linker (KFL) fragment encompassing residues 700–841 ($KFL_{700–841}$) bound strongly to the beads, whereas the shorter fragment $KFL_{700–793}$ was not retained. The absence of $Ca^{2+}$ markedly, but not completely, diminished the association between the PYK2 constructs and CaM (Fig. 1b, c). Uncropped images for all western blots are shown as Supplementary Fig. 1b–d.

These results suggested that both the FERM domain and $KFL_{700-841}$ can mediate CaM binding. To test whether these interactions are direct, we repeated the same CaM agarose bead assay with purified recombinant proteins. In addition to PYK2 FERM, we used the PYK2 $KFL_{728–839}$ fragment [cropped to eliminate the proline-rich motif 2 (PR2) situated between residues 701 and 727] and included the FAK FERM domain as a negative control. Circular dichroism (CD) and differential-scanning fluorimetry (DSF) confirmed that the recombinant PYK2 and FAK FERM domains were correctly folded under the conditions used (Supplementary Fig. 2a–c). Conversely to the cell lysate-based assay, only PYK2 $KFL_{728–839}$ was retained by the CaM beads, but not the PYK2 FERM domain (Fig. 1d).

We reasoned that the FERM domain-containing PYK2 fragments may have been retained in the cell lysate-based assay via indirect interactions through endogenous CaM-binding proteins. For example, Src, a prominent ligand of both PYK2 and FAK, was also retained in the CaM bead assay performed with cell lysates or with purified protein in vitro (Supplementary Fig. 2d). Notably, FAK bound to CaM beads even better than PYK2 in the cell lysate-based assay, highlighting the occurrence of indirect CaM associations in this setting (Supplementary Fig. 2e, f). For this study, we decided to focus on the direct and PYK2-specific association between KFL and CaM.

### PYK2 KFL forms a disordered composite motif for CaM-binding. 
The KFL is the least conserved region between FAK and PYK2 (Fig. 1a), making it a prime candidate for explaining the differential behaviour of the two related proteins. Based on our alignment of FAK and PYK2 sequences, we identified a relatively well-conserved KFL region that was bioinformatically predicted to be disordered, except for an α-helix (residues 790–828 and 807–824 in PYK2 and FAK, respectively; Supplementary Figs. 3 and 4a, b, and alphafold.ebi.ac.uk/entry/Q14289). Sequence analysis suggested that the region surrounding this helix contains putative CaM-binding motifs in PYK2 but not in FAK (Supplementary Fig. 4a, b). We designed and recombinantly produced several constructs comprising the putative helical regions of PYK2 and FAK (Fig. 2a). Of these, FAK $KFL_{764-845}$ was insoluble

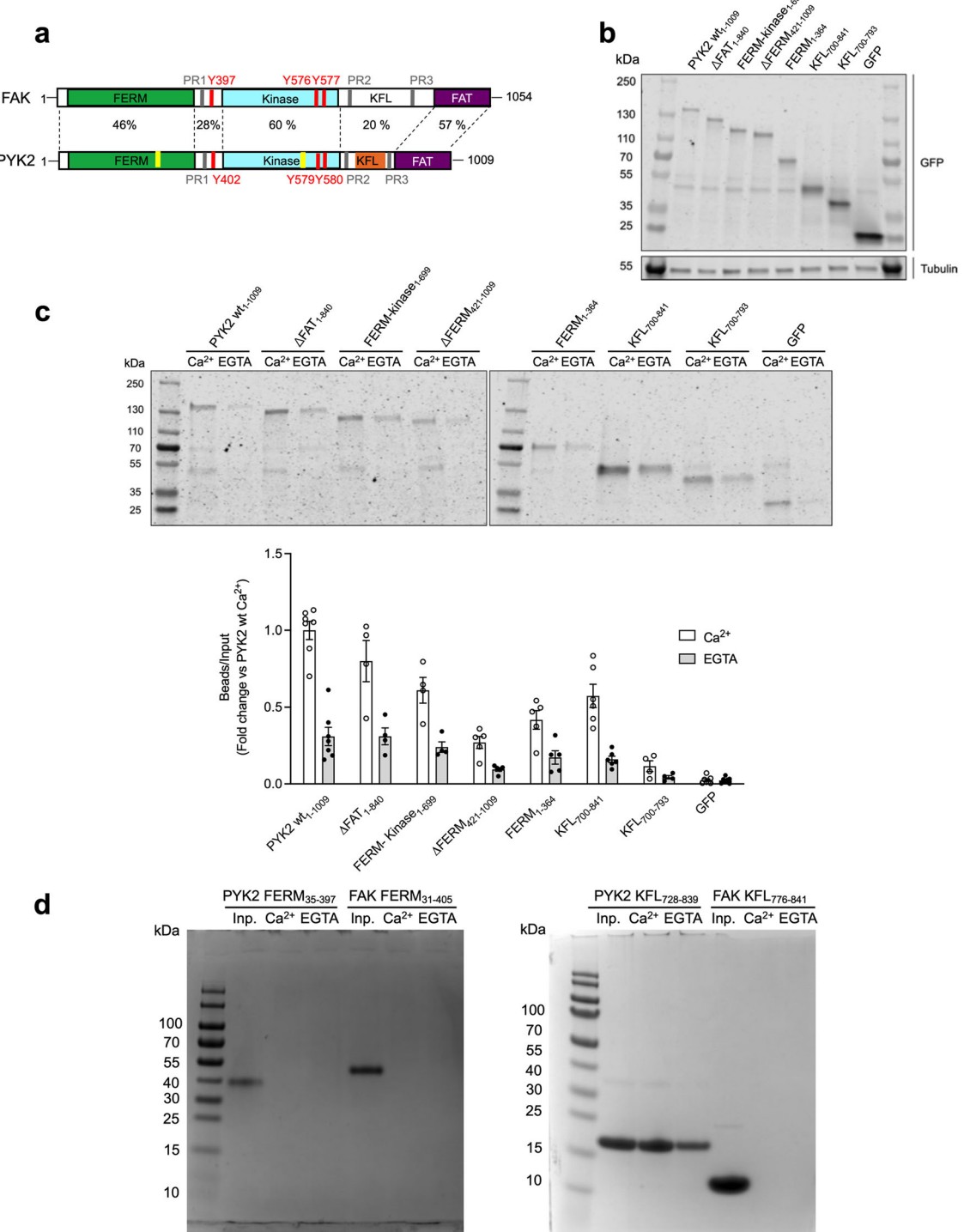

**Fig. 1 Identification of the CaM-binding site in PYK2. a** Structural domain arrangement of FAK and PYK2. The FERM, kinase and FAT domains are colour-coded. Flexible linker regions are shown in white. Tyrosines important for kinase activation through phosphorylation are indicated in red. Proline-rich (PR) motifs 1–3 are marked in grey. Previously suggested $Ca^{2+}$/CaM-binding sites in PYK2 are shown in yellow, and the site determined in this study is in orange. **b** Representative immunoblot of input fractions of the indicated GFP-tagged PYK2 constructs immunolabelled with GFP and tubulin antibodies. **c** *Top:* Representative GFP immunoblot of GFP-PYK2 constructs associated with the CaM Sepharose beads labelled as in **b**. *Bottom:* Graphical representation of GFP immunoblot densitometry associated with beads normalised by corresponding input, presented as foldchange using PYK2 WT $Ca^{2+}$ bead fractions as a reference. Bars correspond to the mean of 4–7 independent experiments, ±SEM. **d** CaM Sepharose bead assay using recombinant purified FERM domains of PYK2 and FAK (left) and KFL regions of PYK2 and FAK (right). For each construct the input (inp.) and the bead fractions were run in adjacent lanes as indicated.

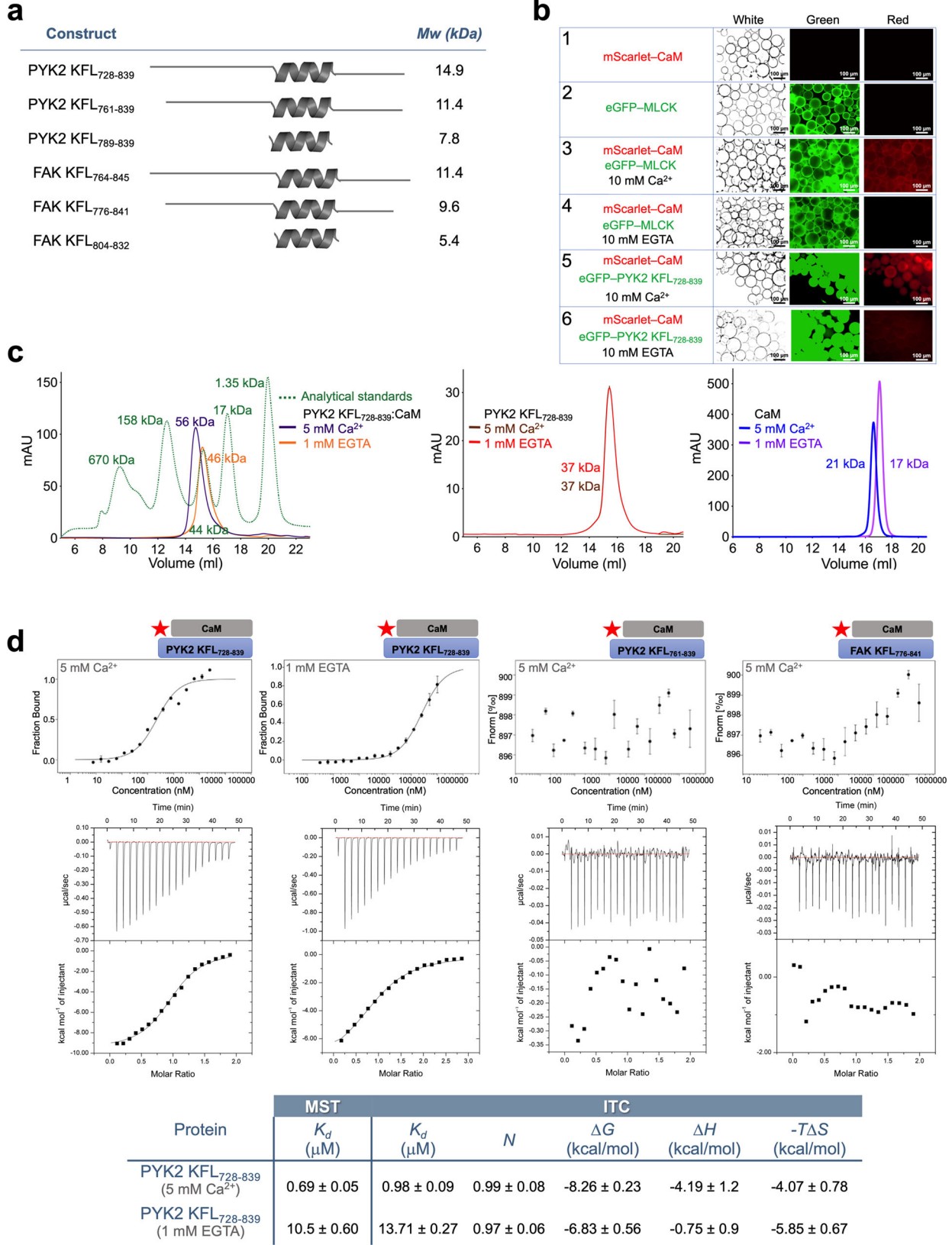

| | MST | ITC | | | | |
|---|---|---|---|---|---|---|
| Protein | $K_d$ (μM) | $K_d$ (μM) | N | $\Delta G$ (kcal/mol) | $\Delta H$ (kcal/mol) | $-T\Delta S$ (kcal/mol) |
| PYK2 KFL$_{728-839}$ (5 mM Ca$^{2+}$) | 0.69 ± 0.05 | 0.98 ± 0.09 | 0.99 ± 0.08 | -8.26 ± 0.23 | -4.19 ± 1.2 | -4.07 ± 0.78 |
| PYK2 KFL$_{728-839}$ (1 mM EGTA) | 10.5 ± 0.60 | 13.71 ± 0.27 | 0.97 ± 0.06 | -6.83 ± 0.56 | -0.75 ± 0.9 | -5.85 ± 0.67 |

and could not be used for in vitro analysis. Using CD, we confirmed that PYK2 KFL$_{728-839}$ and FAK KFL$_{776-841}$ were mostly unstructured with an α-helical content similar to the ones predicted bioinformatically (Supplementary Fig. 5a–e and Supplementary Table 1).

Next, we performed additional assays to corroborate and dissect the association between PYK2 KFL$_{728-839}$ and CaM using recombinant proteins. Our visible immunoprecipitation (VIP) assay and analytical size-exclusion chromatography (SEC) experiments demonstrated that PYK2 KFL$_{728-839}$ associated with Ca$^{2+}$/

**Fig. 2 Analysis of the CaM-binding properties of PYK2 KFL. a** Constructs designed for in vitro characterisation of the KFL region in FAK and PYK2. All constructs were expressed and purified in three different variants, namely with a 6xHis, an eGFP or an MBP tag. **b** Visible immunoprecipitation (VIP) assay. eGFP-fused proteins (green) were bound to GFP–nanobody beads. The CaM-interacting fragment of the myosin light chain kinase (MLCK) was used as positive control. Beads were incubated with mScarlet-fused CaM (mScarlet-CaM), washed and then visualised using white light (White), green light (Green, 488 nm, to visualise eGFP), or red light (Red, 569 nm, to visualise mScarlet). Scale bars are indicated on each panel. **c** *Left*—SEC elution profile showing that PYK2 KFL$_{728-839}$ bound to CaM in the presence (purple) as well as absence (orange) of calcium. The molecular standards are shown in green. The calculated Mw of CaM is 18.5 kDa. *Middle and right* – SEC elution profiles for purified PYK2 KFL$_{728-839}$ or CaM with either 5 mM Ca$^{2+}$ or 1 mM EGTA. **d** *Top*—MST-binding assays. Red star indicates that CaM was fluorescently labelled. *Middle*—ITC data corresponding to the MST interactions, however, with unlabelled CaM injected from the syringe and KFL constructs in the cell. Measured heats (top) and integrated heats (bottom) are shown. *Bottom*— Summary of MST and ITC results. $K_d$: dissociation constant; $N$: stoichiometry; $\Delta G$: change in Gibb's free energy; $\Delta H$: change in enthalpy; $T\Delta S$: change in entropy * temperature in kelvin. Binding experiments are represented as (mean ± SD, $n = 3$).

CaM (Fig. 2b, c). Using MST and ITC, we found that PYK2 KFL$_{728-839}$ bound to Ca$^{2+}$/CaM with a micromolar dissociation constant ($K_d$). Conversely, the PYK2 FERM–kinase or FERM constructs failed to show binding to Ca$^{2+}$/CaM (Supplementary Fig. 5f–h), and PYK2 KFL$_{728-839}$ did not bind to Ca$^{2+}$ in the absence of CaM with measurable affinity by ITC (Supplementary Fig. 5i). Neither PYK2 KFL constructs with shorter flanking regions nor FAK KFL constructs associated with Ca$^{2+}$/CaM in MST or ITC experiments. As expected, FAK FERM did not bind purified CaM in these experiments (Fig. 2d and Supplementary Fig. 5g).

VIP suggested a residual PYK2–CaM association even without Ca$^{2+}$ (Fig. 2b, row 6). ITC and MST confirmed that CaM bound to PYK2 KFL$_{728-839}$ in the absence of Ca$^{2+}$, albeit with a 10-fold weaker affinity (Fig. 2d). Complex formation without Ca$^{2+}$ was further corroborated by DSF showing that the presence of PYK2 KFL$_{728-839}$ increased the melting temperature $Tm$ of CaM by 4 °C in the absence of Ca$^{2+}$ (Supplementary Fig. 6a), whereas PYK2 KFL$_{728-839}$ itself did not show an unfolding transition between 25 °C and 95 °C (Supplementary Fig. 6a).

With or without Ca$^{2+}$, ITC revealed a stoichiometry $N$ of ~1 for the PYK2 KFL$_{728-839}$–CaM complex. However, with or without Ca$^{2+}$, CD analysis showed that the secondary structure content of the complex was not higher than the sum of the proteins measured individually (Supplementary Fig. 5a, b and Supplementary Table 1). Notably, synthesised peptides covering only the bioinformatically identified canonical CaM-binding motifs from PYK2 KFL$_{728-839}$ did not bind to CaM, showing that the interaction requires a larger KFL region (Supplementary Fig. 6b). Taken together, these results indicated that PYK2 KFL$_{728-839}$ harbours an unusually long and largely disordered CaM-binding motif that retains low affinity in the absence of Ca$^{2+}$ and does not gain secondary structure upon binding.

**PYK2 KFL forms disordered dimers.** In our analytical SEC analysis (Fig. 2c, middle panel) PYK2 KFL$_{728-839}$ eluted with a molecular weight markedly greater than expected for a monomer, suggesting self-association. Indeed, SEC with multiangle light scattering detection (SEC-MALS) (Fig. 3a, b and Supplementary Fig. 7a) and analytical ultracentrifugation (AUC) (Supplementary Fig. 7b) revealed mass and size estimates corresponding to dimers for those KFL constructs that contained large regions flanking the predicted helix, namely PYK2 KFL$_{728-839}$ and FAK KFL$_{776-841}$. Conversely, the KFL fragments with short flanking regions did not self-associate (Fig. 3a, b and Supplementary Fig. 7a). MST found that the dimerisation $K_d$ for both PYK2 KFL$_{728-839}$ and FAK KFL$_{776-841}$ was ~0.8 μM and confirmed that short fragments did not self-associate (Fig. 3c, d and Supplementary Fig. 7c, d; also refer to Momin et al.[43]). Fluorescence anisotropy experiments validated these self-association $K_d$s (Fig. 3e, f, g and Supplementary Fig. 7e, f). However, PYK2 KFL$_{728-839}$ and FAK KFL$_{776-841}$ did not bind each other in our MST experiments (Supplementary Fig. 7g), showing that the dimerisation is specific.

Our SEC small-angle X-ray scattering analysis of PYK2 KFL$_{728-839}$ alone and fused to the maltose-binding protein (MBP) also identified a molecular mass corresponding to a dimeric species for both constructs (Supplementary Fig. 7h). The radius of gyration of PYK2 KFL$_{728-839}$ (without MBP) was larger than that expected for a dimeric globular protein but smaller than that of a completely disordered protein[44], indicating an elongated shape and/or significant structural flexibility (Supplementary Table 2). The sigmoidal shape of the Kratky plot corroborated that PYK2 KFL$_{728-839}$ has a high degree of structural flexibility despite being mostly dimeric under these experimental conditions (Supplementary Fig. 7i).

Next, we used nuclear magnetic resonance (NMR) to obtain residue-specific information for this unusually flexible dimerisation module of PYK2. We expressed and purified [$^{13}$C,$^{15}$N] PYK2 KFL$_{728-839}$ and assigned ~90% of all resonances (Fig. 4a). Secondary structure assignment based on NMR chemical shift analysis revealed an α-helical segment between residues 791–812, which is a little shorter than that predicted from the sequence and calculated from experimental CD data (Supplementary Fig. 8a). The salient features of two-dimensional (2D) [$^1$H-$^{15}$N] hetero-nuclear single quantum coherence (HSQC) experiments agreed with a highly flexible structure. The spread of the chemical shifts was limited between 7.7 and 8.7 ppm on the $^1$H axis. Moreover, the chemical shifts of the side-chain $^{15}$N/$^1$H of both tryptophans were located in a region associated with solvent exposure (10.13 ± 0.02 ppm and 129.95 ± 0.35 ppm), indicating these tryptophans are not involved in local hydrophobic clusters (Fig. 4a).

Dilution of [$^{13}$C,$^{15}$N] PYK2 KFL$_{728-839}$ from 100 to 10 μM, monitored using the 2D [$^1$H-$^{15}$N] HSQC spectra, revealed selective peak broadening and chemical shift perturbations (CSPs) as the percentage of PYK2 KFL$_{728-839}$ molecules in the dimer decreased from 94% to 82%, in accordance with the dimerisation $K_d$ of ~0.8 μM (Fig. 4b, top. Supplementary Fig. 9). The regions of significant CSP changes broadly correlated with an analysis of the peak intensity change as a result of sample dilution (Fig. 4b, bottom). Both methods revealed that the dilution affected many regions relatively weakly, and that these regions were scattered across the length of the molecule. We concluded that PYK2 KFL$_{728-839}$ dimerisation is a fuzzy and rather unspecific process.

The 17 significantly altered backbone amide signals (>inter-quartile range of CSPs) corresponded to residues that were evenly scattered along PYK2 KFL$_{728-839}$ (Fig. 4b, c and Supplementary Table 3). These 17 residues did not display a clear physicochemical bias; seven were polar, six were hydrophobic, two were positively charged and one was negatively charged. Only residues K803, M804, I807, and L808 formed a connected, predominantly hydrophobic surface patch on the helical region, suggesting the particular importance of these residues for dimerisation (Fig. 4c).

To address the possibility that an intermediate exchange regime concealed the resonances of the bound state at the

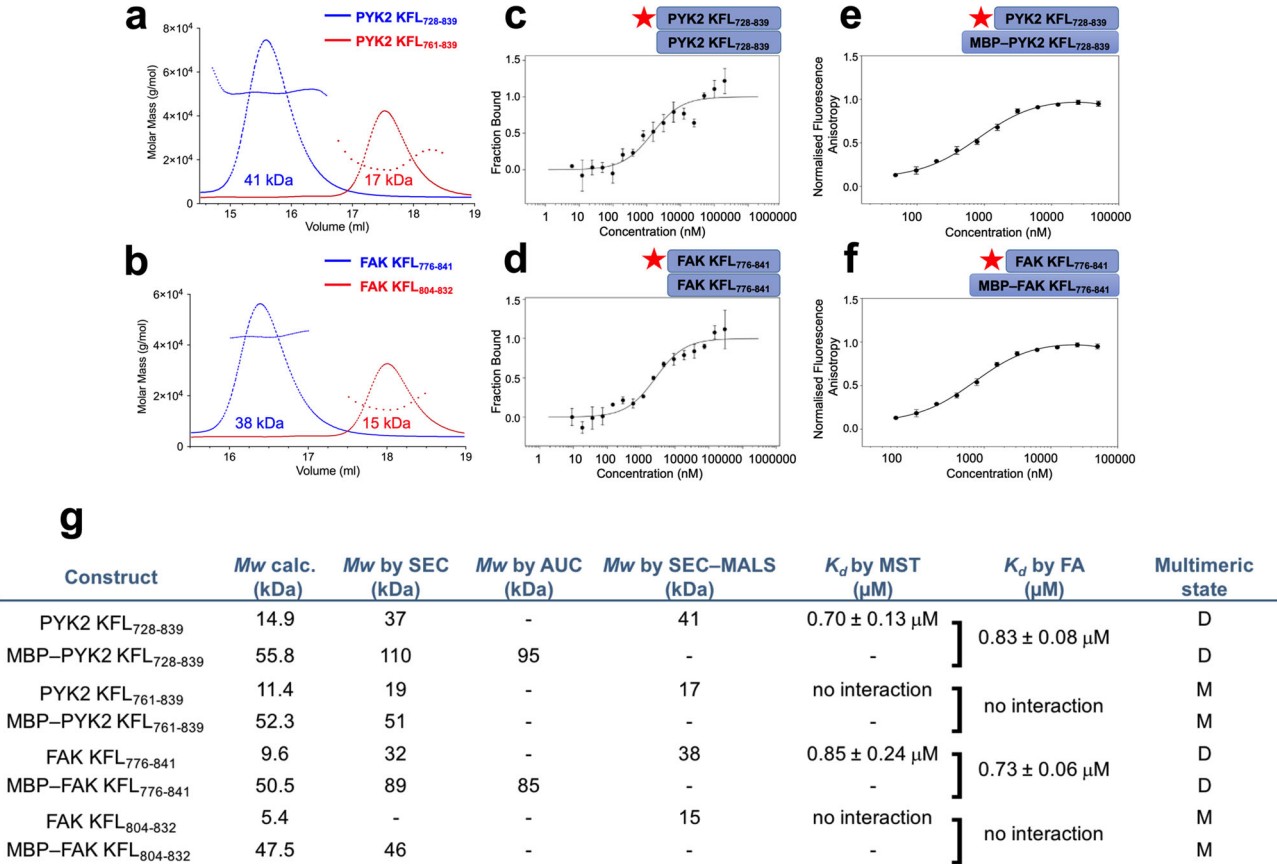

**Fig. 3 PYK2 and FAK KFL dimerise. a, b** SEC-MALS elution profiles on purified proteins. The *Mw* deduced for each peak is given. The additional lines show the molar mass distribution. **c, d** MST and **e, f** fluorescence anisotropy (FA) studies. The red star indicates the fluorescently labelled protein. MBP fusions were used to increase the *Mw* of the unlabelled molecule in FA. **g** Summary table, also including data derived from Supplementary Fig. 7a–f). The single square bracket indicates that the $K_d$ in FA was established between fluorescently labelled protein and MBP-fused unlabelled protein. D: dimer; M: monomer. Binding experiments are represented as (mean ± SD, $n = 3$).

700 MHz frequency used, we also recorded the 100 µM [$^{15}$N]-PYK2 KFL$_{728-839}$ sample at 950 MHz. We neither observed new signals appearing, nor a significant narrowing of signals at 950 MHz, nor significant modulations in the peak intensity ratio between measurements at 700 and 950 MHz, supporting that our analysis was not affected by a significant contribution of intermediate exchange (Fig. 4d and Supplementary Fig. 10).

In further support of the dynamic and fuzzy dimerisation mode, $^1$H/$^{13}$C chemical shifts assigned for the residues of the same type, including side-chain methyl groups, were similar. In agreement with the modest signal dispersion on 2D [$^1$H-$^{13}$C] and [$^1$H-$^{15}$N] HSQC spectra, we detected only sequential $i + 1,3,4$ nuclear Overhauser effects (NOEs) within the helix and $i \pm 1$ NOEs elsewhere from the 3D $^{13}$C- and $^{15}$N-edited NOE spectroscopy (NOESY)–HSQC spectra (Supplementary Fig. 8b). Finally, we did not observe any intermolecular NOEs in $^{13}$C/$^{15}$N-filtered NOE experiments performed on a mixture of [$^{13}$C-$^{15}$N]-labelled and unlabelled PYK2 KFL$_{728-839}$ at two pH values (6.5 and 7.5). Taken together, these observations confirm the fuzzy dimerisation mode of PYK2 KFL$_{728-839}$, in which several structural conformations and inter-chain contacts are constantly in a dynamic exchange (Supplementary Fig. 8c).

**PYK2-KFL binds to CaM in a process involving more than one linear peptide association.** As a next step, we used NMR to map the interaction sites between PYK2 KFL$_{728-839}$ and CaM. We titrated [$^{13}$C,$^{15}$N]-PYK2 KFL$_{728-839}$ with unlabelled CaM and, in

a second series, assigned [$^{13}$C,$^{15}$N]-CaM (Supplementary Fig. 11a, b) and titrated it with unlabelled PYK2 KFL$_{728-839}$. In agreement with the results of our other binding experiments, we observed an interaction between PYK2 KFL$_{728-839}$ and CaM in the absence and the presence of Ca$^{2+}$ (Fig. 5a–d and Supplementary Fig. 11c–f). In the Ca$^{2+}$-free association, twice as many resonances from the CaM C-terminal lobe (16 residues) than from the N-terminal lobe (7 residues) significantly broadened beyond the detection or shifted their $^1$H-$^{15}$N resonances (Supplementary Table 4). In PYK2 KFL$_{728-839}$, most of the affected residues clustered within the helical region (residues 791–812) and flanking residues (R790, K813, M815; Fig. 5a, b). For both CaM and PYK2 KFL$_{728-839}$, the Ca$^{2+}$-free association prominently involved charged residues, with good overall charge complementarity between the nine negative charges from CaM (seven from its C-lobe) and seven positive charges from PYK2 KFL$_{728-839}$. Additional marked changes occurred in resonances from hydrophobic residues (CaM: 9; PYK2 KFL$_{728-839}$: 14) (Fig. 5a, b, e, f and Supplementary Table 4).

In the presence of Ca$^{2+}$, CaM [$^1$H-$^{15}$N] backbone amide resonances were lost due to line broadening starting from the Ca$^{2+}$/CaM:PYK2 KFL$_{728-839}$ ratio of 1:0.5. All CaM resonances disappeared at the ratio of 1:4, as expected for the >70 kDa protein complex resulting from a 2:2 interaction between Ca$^{2+}$/CaM and PYK2 KFL$_{728-839}$ (Fig. 5c, d and Supplementary Fig. 11f). We identified the most strongly interacting residues in CaM as those in which the resonances broadened out or decreased below a threshold at Ca$^{2+}$/CaM:PYK2 KFL$_{728-839}$

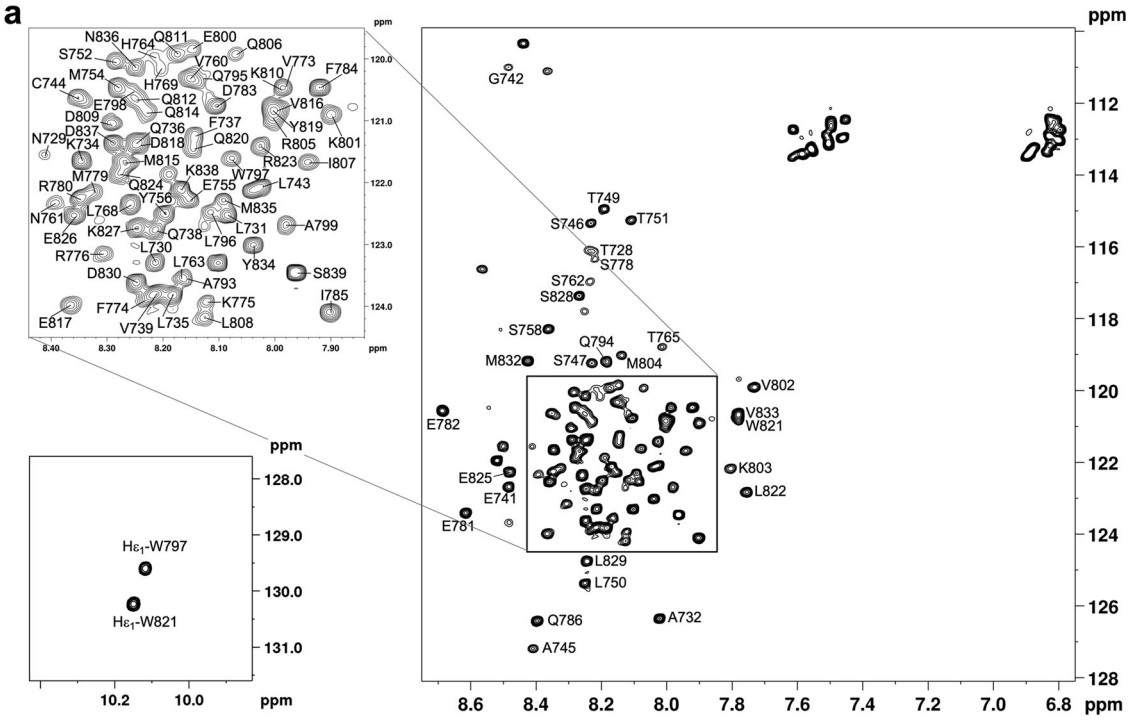

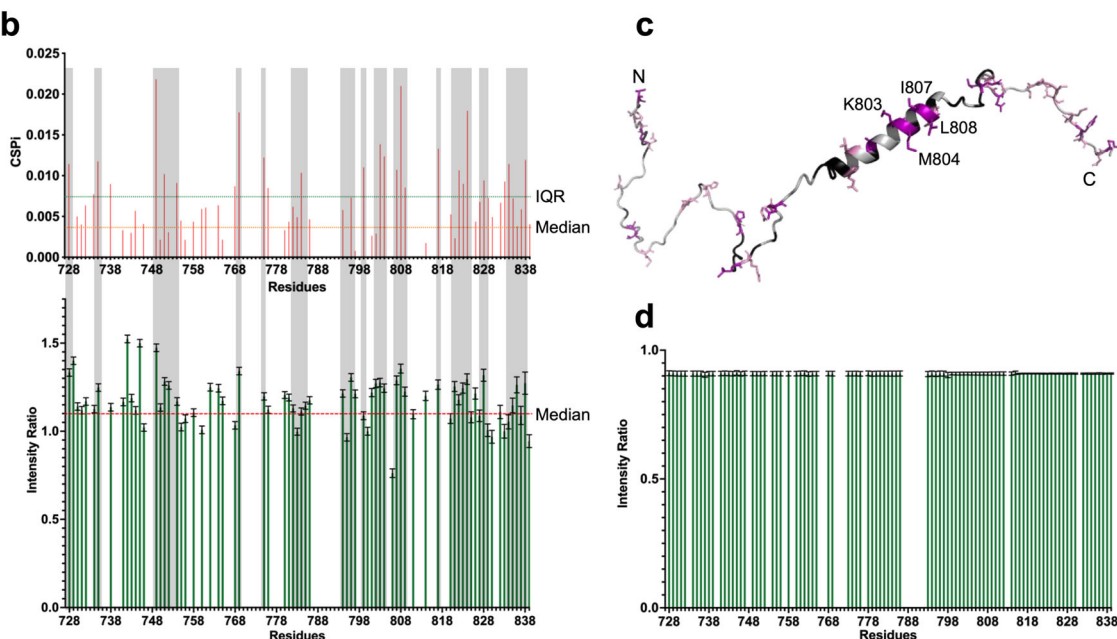

**Fig. 4 Structural characterisation of PYK2 KFL728-839 by NMR. a** [¹H-¹⁵N] HSQC spectrum of PYK2 KFL$_{728-839}$ recorded on a Bruker Avance III 950 MHz spectrometer at 25 °C, pH 6.5, on a 250 μM ¹⁵N-uniformly labelled sample. Cross-peak assignments are indicated using the one-letter amino acid and number code. The central part of the spectrum is expanded in the insert at the top left. Assignment of tryptophan indole NH proton resonances is represented at the bottom left. **b** (*Top*) CSP occurring as a result of diluting [¹³C,¹⁵N] PYK2 KFL$_{728-839}$ from 100 μM (8 scans) to 10 μM (128 scans), monitored by 2D [¹H-¹⁵N] HSQC. The orange horizontal line indicates the median threshold for minor shifts and the horizontal green line and shows the interquartile range (IQR) threshold for major shifts. (*Bottom*) Plot of 2D [¹H-¹⁵N] HSQC intensity ratios of [¹³C,¹⁵N] PYK2 KFL$_{728-839}$. The ratios were calculated as 10 μM (128 scans) divided by 100 μM (8 scans). The intensity value at 100 μM was taken as a reference and used to normalise the intensity value at 10 μM to compensate for an overall loss of intensity upon dilution. Red dotted line indicates the median. Grey-shaded zones indicate residue regions where significant CSPs correspond to regions of lower relative intensity (**c**) CSPs from **b** were mapped on a representative 3D structure of PYK2 KFL$_{728-839}$. Major shifts are marked in magenta, minor shifts are coloured in pink and prolines, overlapping and unassigned residues are marked in black. N and C indicate the N-terminal and C-terminal of the molecule. **d** Plot of 2D [¹H-¹⁵N] HSQC intensity ratios of [¹H,¹⁵N] PYK2 KFL$_{728-839}$. The ratios were calculated as 100 μM (8 scans) measured on 700 MHz divided by 100 μM (8 scans) measured on 950 MHz.

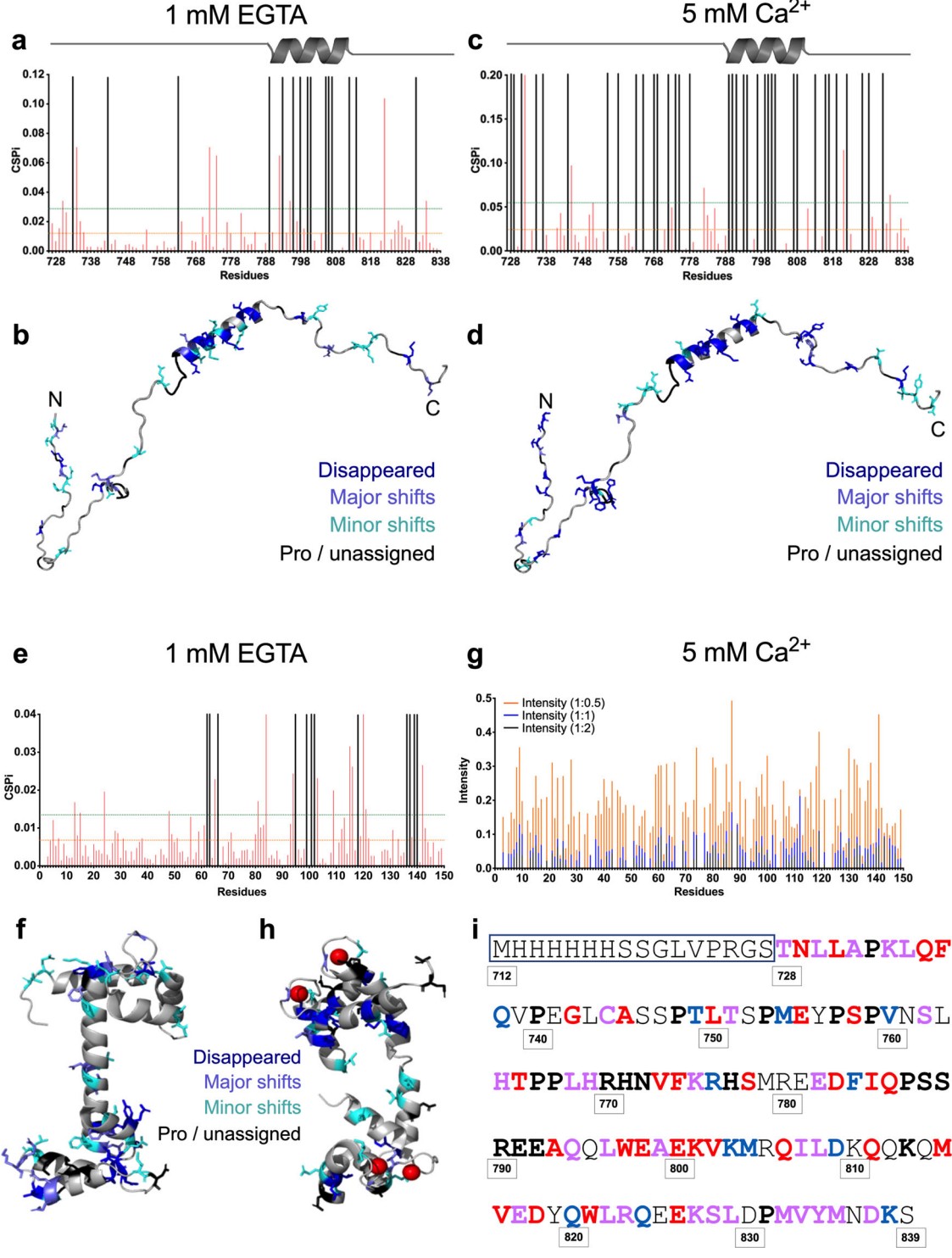

ratios up to 1:1. At the 250 μM concentration used for each of the proteins, and given the association $K_d$ of ~1 μM, ~96% of the labelled $Ca^{2+}$/CaM was bound to PYK2 KFL$_{728-839}$. Although the mapping cannot be done with high confidence based on these data, considerably more hydrophobic CaM residues appear to be involved compared with the findings for titrations without $Ca^{2+}$. Those residues were mostly contributed by the CaM N-lobe (Fig. 5g, h).

With $Ca^{2+}$, the imprint of unlabelled $Ca^{2+}$/CaM was also spread more broadly across [$^{13}$C,$^{15}$N]-PYK2 KFL$_{728-839}$.

However, even at the highest $Ca^{2+}$/CaM concentration, line broadening was not as severe as that observed for [$^{13}$C,$^{15}$N]-CaM titrated with PYK2 KFL$_{728-839}$, supporting that PYK2 KFL$_{728-839}$ retains significant flexibility when bound to $Ca^{2+}$/CaM (Supplementary Fig. 11d). The presence of $Ca^{2+}$ markedly increased the involvement of hydrophobic, polar, and acidic residues in PYK2 KFL$_{728-839}$. These additional residues were predominantly located within the regions outside the KFL helix, while the contributions from the helix region remained comparable in titrations with or without $Ca^{2+}$ (Fig. 5c, d and Supplementary Table 4).

**Fig. 5 NMR mapping of the residues contributing to the CaM–PYK2 KFL association.** CSP analyses of 150 μM [$^{13}$C,$^{15}$N] PYK2 KFL$_{728-839}$ titrated with CaM in the **a** absence and **c** presence of Ca$^{2+}$. Orange and green horizontal lines indicate the threshold for major shifts (Δppm + 2 std) and minor shifts (Δppm + 1 std), respectively. Resonances that disappeared are indicated in black. CSPs from **a** and **c** are mapped onto a structural representative of PYK2 KFL$_{728-839}$ in **b** and **d**, respectively. Dark blue: residues for which peaks disappeared; slate blue: major shifts; cyan: minor shifts; black: prolines and unassigned residues. N and C indicate the N-terminal and C-terminal of the molecule. **e** CSP analysis of 150 μM [$^{13}$C,$^{15}$N] apo-CaM titrated with PYK2 KFL$_{728-839}$. Colouring according to **a**. **f** Mapping of the data from **e** onto the 3D structure of apo-CaM (PDB ID 4e53). Colouring according to **b**. **g** Plot of peak intensities recorded for 150 μM [$^{13}$C,$^{15}$N] Ca$^{2+}$/CaM titrated with PYK2 KFL$_{728-839}$ at 10 °C. The intensity value without the binding partner (corresponding to a ratio of 1:0) is taken as a reference and used to normalise the intensity values of subsequent titrations (1:0.5, 1:1, 1:2) to compensate for an overall loss of intensity as the binding partner is added to the solution at increasing concentrations. Intensities per residue are colour-coded by titration ratio as indicated. **h** Intensity changes from **g** are mapped onto the 3D structure of CaM (PDB ID 1 × 02) according to dark blue: residues that disappeared at ratio 1:0.5; slate blue: residues with an intensity less than 0.045 at a ratio of 1:0.5; cyan: residues that have an intensity more than 0.045 at 1:0.5 but disappear at 1:1; black: prolines and unassigned residues. Red spheres represent Ca$^{2+}$. **i** Sequence of the PYK2 KFL$_{728-839}$ construct used. The non-natural 6xHis-tag is boxed in blue. Pink residues indicate residues identified as contributing to both dimerisation and CaM binding. Blue residues are those assumed contributing only to dimerisation (based on CSPs), and red residues are those that only contribute to CaM binding (in the absence or presence of Ca$^{2+}$). Prolines and unassigned residues are marked in bold black. Note that the confidence of the residue mapping is low because of the fuzzy nature of the binding events.

We concluded that without Ca$^{2+}$, CaM and PYK2 KFL$_{728-839}$ form a charge-dominated association, largely based on the CaM C-lobe and PYK2 KFL$_{728-839}$ helix region. The addition of Ca$^{2+}$ promotes more hydrophobic and polar interactions between additional regions on both proteins while largely, but not entirely, preserving the interactions occurring without Ca$^{2+}$. These additional interactions enhance the binding affinity and markedly constrain the dynamics of CaM, but not of PYK2 KFL$_{728-839}$, as judged from line broadening.

The effect of Ca$^{2+}$ on the association is in agreement with the previously reported mechanism that Ca$^{2+}$ leads to structural changes in CaM that expose more hydrophobic residues, enabling stronger ligand binding[45]. Hence, the ligands that bind to CaM in the absence of Ca$^{2+}$ tend to form a weaker charge-based association, particularly with the CaM C-lobe[45]. We noted that the CaM regions involved in PYK2 KFL$_{728-839}$ binding in the absence of Ca$^{2+}$ are reminiscent of those mediating Ca$^{2+}$-independent associations with an IQ motif (named after the first two amino acids commonly found in this consensus)[46] (Supplementary Fig. 11g). However, as shown above, the IQ motif alone did not bind to CaM, demonstrating that the association between CaM and PYK2 KFL$_{728-839}$ was more complex, in line with our NMR-binding site mapping. Finally, we observed a marked overlap between the residues identified as contributing to dimerisation and those contributing to CaM binding, suggesting that both events are functionally linked. Noteworthy exceptions were both tryptophans (W797 and W821) that were only involved in CaM binding (Fig. 5i). Among FAK and PYK2 sequences, the residues involved in PYK2 KFL dimerisation are more conserved than those implicated in CaM binding, in agreement with dimerisation being a conserved FAK/PYK2 function and CaM binding being unique to PYK2 (conservations scores are 0.65 and 0.53, respectively, while the average conservation score of the KFL region is 0.56, where 0 is not conserved and 1 is completely conserved).

**PYK2 control through the fuzzy dimeric CaM-binding element.** We designed a series of experiments to probe the functional implications of our results. Given that Ca$^{2+}$ influx promotes PYK2 self-association and Y402 autophosphorylation in cells[9,20,21], we tested whether there was synergy between CaM binding, PYK2 KFL dimerisation, and PYK2 autophosphorylation. Previous studies have provided evidence that PYK2 autophosphorylation, in response to G protein-coupled receptor stimulation, is prevented by CaM antagonists[47,48]. Accordingly, we found that the CaM antagonist calmidazolium blocked the membrane depolarisation-induced increase of PYK2

autophosphorylation in PC12 cells (Fig. 6a). However, in vitro, we observed only a mild increase (~2.5-fold) in the dimerisation strength of PYK2 KFL$_{728-839}$ in the presence of Ca$^{2+}$/CaM compared with the dimerisation strength of PYK2 KFL$_{728-839}$ alone (Fig. 6b). The weak effect of Ca$^{2+}$/CaM binding on PYK2 KFL$_{728-839}$ dimerisation suggests the involvement of other factors in cellular full-length PYK2.

FAK and PYK2 self-associate only under specific cell conditions and in particular subcellular locations[20–23]. Therefore, we examined whether, in the PYK2 monomer, KFL$_{728-839}$ interacts with other regions of the same PYK2 polypeptide. Our ITC and MST experiments revealed that monomeric PYK2 KFL$_{728-839}$ failed to bind to the FERM or FAT domains (Supplementary Fig. 12a). However, monomeric PYK2 KFL$_{728-839}$ bound to the PYK2 FERM–kinase fragment with micromolar affinity (Fig. 6c). These observations suggest that when PYK2 is monomeric, PYK2 KFL$_{728-839}$ associates with the FERM–kinase fragment. This association may conceal the dimerisation and CaM-binding surface of the KFL as well as possible flanking motifs that mediate ligand interaction (e.g., PR2 and PR3) or phosphorylation (Fig. 6d).

**Discussion**

The hallmark difference between PYK2 and FAK is the capability of the former to sense and respond to increases in cellular Ca$^{2+}$. Although there is a general agreement that the major (but not the only) mechanism for Ca$^{2+}$ sensing is based on the recognition of Ca$^{2+}$/CaM, its molecular mechanism remains a matter of debate[16,40–42]. In this study, we demonstrated that Ca$^{2+}$ can control PYK2 by reinforcing an association between CaM and a PYK2 linker region located between its kinase and FAT domains. This region, PYK2 KFL$_{728-839}$, constitutes an unusual CaM-binding motif. It contains a short amphipathic helix and several putative CaM-interacting peptide motifs. Individually, however, these elements are not enough; it is their combination that creates a fuzzy, disordered CaM-binding module where multiple dispersed KFL residues transiently touch various CaM sites. Although it specifically associates with CaM in a 1:1 stoichiometry, PYK2 KFL$_{728-839}$ largely retains its flexibility during the interaction. This module binds CaM already weakly in the absence of Ca$^{2+}$, and the structural CaM rearrangement induced by Ca$^{2+}$ creates additional sites that extend and tighten the association.

The preservation of disorder in the association of unstructured proteins with their ligands has been observed previously[49], and several ligands were described to form "fuzzy" complexes with CaM. However, almost all of these CaM-interacting proteins still

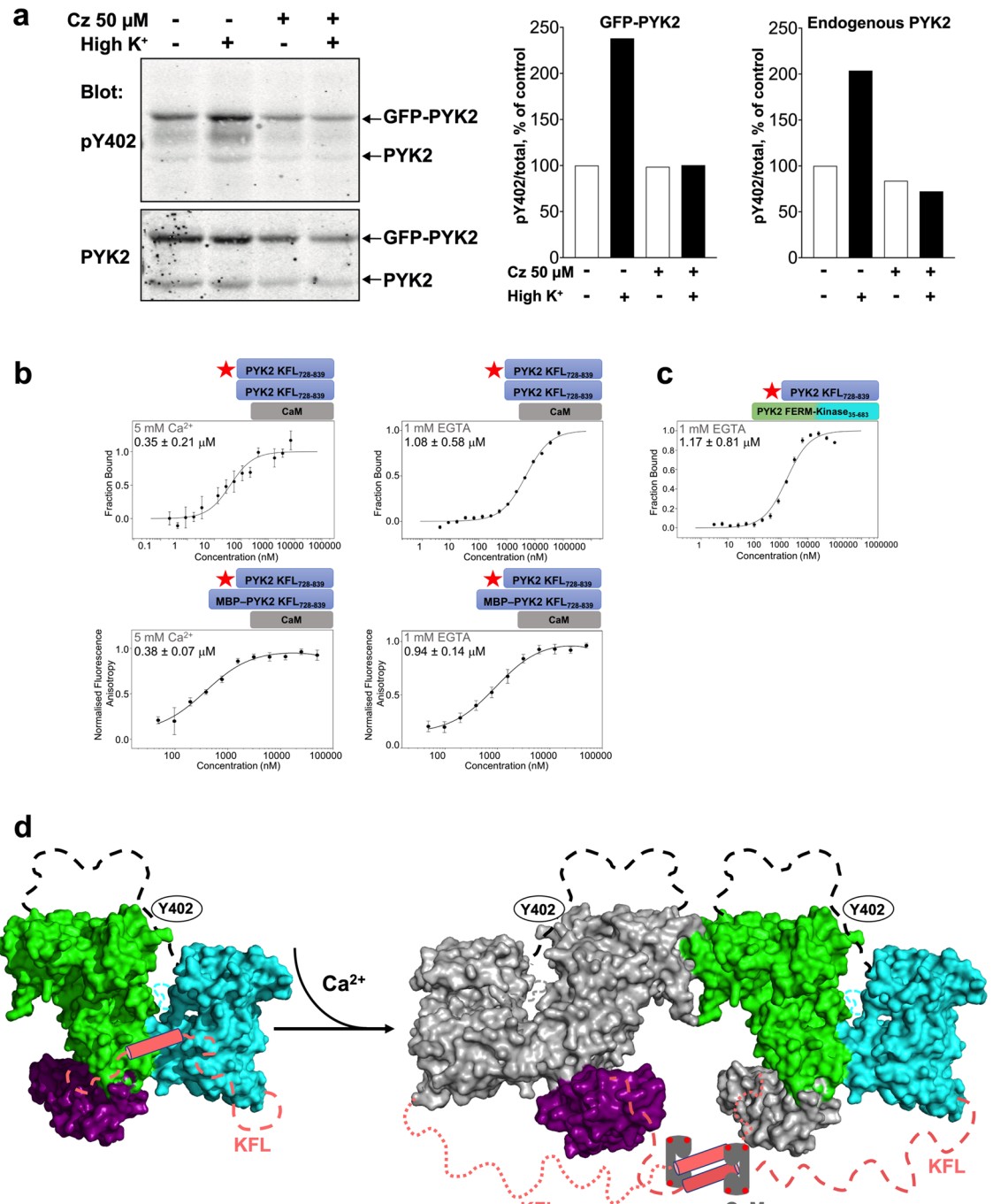

**Fig. 6 Control of PYK2 function through fuzzy CaM-binding and dimerisation of KFL. a** CaM antagonist calmidazolium (Cz, 50 μM) blocks PYK2 Y402 phosphorylation induced by membrane depolarisation in PC12 cells. Depolarisation was induced by isosmotic replacement of 40 mM NaCl by 40 mM KCl in the extracellular medium for 3 min. Graphical representation of PYK2 pY402 over total PYK 2 densitometry for transfected and endogenous PYK2 bands (mean values of 2 replicates). **b** Dimerisation of PYK2-KFL$_{728-839}$ in the presence of 10 μM CaM. Experiments were carried out by MST (*top*) and FA (*bottom*) in the presence of Ca$^{2+}$ or EGTA. The fluorescently labelled protein (indicated by a red star) was at a concentration of 50 nM, and the unlabelled protein was serially diluted from a concentration of 100 μM. The condition (Ca$^{2+}$ or EGTA), and the derived $K_d$ values are noted in the figures. **c** MST analysis of fluorescently labelled PYK2 KFL$_{728-839}$ was kept at 50 nM (where it is predominantly monomeric) and titrated with unlabelled PYK2 FERM-kinase. The presence of Ca$^{2+}$ or EGTA and the $K_d$ are stated. Binding experiments are represented as (mean ± SD, $n = 3$). **d** Proposed molecular mechanism. *Left:* Theoretical model of the closed monomeric PYK2, adopted in the absence of Ca$^{2+}$ and other specific stimuli. Molecular surfaces of the FERM (green), kinase (cyan) and FAT (magenta) domains are shown. The FERM–kinase association was modelled based on the corresponding FAK fragment (PDB accession 2j0k). PYK2 FAT was docked onto the FERM domain as modelled for FAK in ref. [23]. The flexible linker regions are shown as dashed lines. The position of Y402 is indicated. *Right:* Theoretical model of the pre-activation Ca$^{2+}$/CaM-bound PYK2 dimer, modelled on PDB accessions 2j0k and 4ny0. CaM is shown as grey rectangle with the four red dots illustrating the bound Ca$^{2+}$ ions. Note that the CaM–KFL complex is highly disordered, and not symmetric as shown in this illustration. Colours as in the left panel, but FERM, kinase and FAT domains of the second molecule are coloured in grey.

contained a well-defined helical region and/or fixed anchoring positions that stably associated with at least one CaM lobe[50,51], even though several binding modes may be possible, depending on the conditions[52]. A closest known match to our system may be the association of a twenty-residue segment of the myelin basic protein (MBP$_{145-165}$) with CaM where a stable binding pose has not been identified[53]. However, MBP$_{145-165}$ is much shorter than PYK2 KFL$_{728-839}$, monomeric, and binds with a more than ten-fold lower affinity and more ambiguous stoichiometry to CaM. Therefore, the association of PYK2 KFL$_{728-839}$ with CaM may present an extreme case within the spectrum of fuzzy CaM complexes.

Moreover, PYK2 KFL$_{728-839}$ forms a disordered dimer. The sub-micromolar dimerisation $K_d$ precluded a full relaxation analysis, and hence a confident assignment of the residues involved in dimerisation. However, CSPs and intensity changes showed that the dimerisation results from many poorly specific associations that are dispersed along the whole PYK2 KFL$_{728-839}$ region. Self-association of disordered regions has already been described. In many cases, however, homo-multimerization is associated with stable long-range contacts[54], transitions to folding (e.g., α-synuclein)[55], MazE[56], papillomavirus E7[57], electrostatic interactions between oppositely charged regions (e.g., HMGA2)[58], and/or defined residue-residues interactions (e.g., c-Myc PEST)[59], PQBP-1 K192Sfs*7[60]. In our case, we demonstrated that the PYK2 KFL$_{728-839}$ dimerises without these features, in a fuzzy and flexible interaction that requires the full ~110 residue sequence. These features are reminiscent of those of the cytoplasmic domain of the T-cell receptor ζ chain or the chaperon 7B2. However, both may show a more promiscuous concentration-dependent oligomerization into dimers, tetramers and higher-order oligomers and are likely to adopt a more structured state upon binding to a client protein[61,62]. Thus, the KFL expands the structural landscape of IDR self-association and may help analysing other IDRs for which homomultimers have been reported, but not fully structurally resolved (e.g., FEZ1)[63], and the N-terminal domain of Usp[64].

How does this disordered interaction module allow Ca$^{2+}$-dependent control of PYK2? In the case of FAK, kinase activation requires dimerisation and possibly higher-order self-association to initiate autophosphorylation in *trans*[22,23]. For this, FERM–FERM, FAT–FERM, and KFL–KFL interactions collaboratively allow FAK to self-associate in a way that can be controlled by multiple factors[17,22]. PYK2 self-association in response to increases in Ca$^{2+}$ levels are well documented in cellular assays[11,28,29], but the molecular basis remains unknown. We noted that in the crystals of the PYK2 FERM domain (PDB 4eku), this domain forms the same dimeric structure as the FAK FERM domain, with W273 being in the equivalent interface position as the dimerisation-critical W266 in FAK[22] (Supplementary Fig. 12b). Hence, we propose that PYK2 dimerisation is based on a synergistic multidomain association akin to that of FAK, where weak FERM–FERM contacts are stabilised by ligand-controlled supplementary interactions. Ca$^{2+}$/CaM is one of these ligands that promotes PYK2 self-association by stabilising the KFL dimer.

Although the presence of Ca$^{2+}$/CaM only increased PYK2 KFL$_{728-839}$ dimerisation two to three-fold in our MST and FA experiments (Fig. 6b), this effect may be reinforced by supplementary factors in a cellular setting. For example, the effect of Ca$^{2+}$/CaM may be enhanced by additional regions within the full PYK2 protein, by destabilising the association of KFL with the FERM–kinase fragment in the monomer, and/or by additional ligands, such as postsynaptic density protein 95 (PSD-95)[21].

Together with the available literature, our study allows us to propose a mechanistic framework for the transition of PYK2 from the monomeric state to the active dimer (Fig. 6d): In the absence of Ca$^{2+}$ (and other specific stimuli), PYK2 is a monomer in which FERM, kinase, KFL, and possibly FAT domains all associate with each other (Fig. 6d, *left panel*). The resulting "close" conformation conceals some or all of the various functional motifs in the KFL region, including protein interaction motifs (PR2 and PR3), subcellular localisation motifs (NTS and NES), (de)phosphorylation sites, and cleavage sites[6,65]. The association of the monomeric KFL with the FERM–kinase fragment may stabilise the kinase in its inactive conformation. We speculate that a weak association occurring between Ca$^{2+}$-free CaM (not shown in Fig. 6d) and the PYK2 KFL$_{728-839}$ could enrich CaM close to PYK2 prior to Ca$^{2+}$ influx, and/or weaken the inhibitory association of the KFL with the FERM–kinase fragment. Thus, the Ca$^{2+}$-free association could help prime the system for a faster response upon Ca$^{2+}$ influx. Ca$^{2+}$ influx enhances the association of Ca$^{2+}$/CaM with PYK2 KFL$_{728-839}$, helping to liberate the linker region from its FERM–kinase association (Fig. 6d, *right panel*). In a subcellular location with high local PYK2 concentration, Ca$^{2+}$/CaM-enforced KFL dimerisation will stabilise the weak FERM–FERM domain interaction, while releasing the FERM–kinase fragment, thereby priming the protein for trans-autophosphorylation. In this dimeric CaM-bound conformation, the functional motifs on the KFL would be accessible, thereby leading to a catalytically active and ligand-accessible Src–PYK2 scaffold.

CaM has already been shown to promote dimerisation in other ligands. For example, one CaM molecule can tether two molecules of the oestrogen receptor α or of the Na$^+$/H$^+$ exchanger NHE1 by simultaneously binding one molecule with each, its N- and C-terminal lobe, forming a 2:1 complex[66,67]. However, conversely to these examples, CaM reinforces the already strong PYK2 KFL$_{728-839}$ dimer in a fuzzy 2:2 association.

Collectively, our results describe a flexible protein module that expands the structural and functional range for CaM binding and self-association. The features identified for this element in PYK2 may allow the discovery of other fuzzy interaction modules in eukaryotic proteomes. Our analysis also shows how PYK2 has adapted the mechanistic framework inherited from FAK for Ca$^{2+}$ sensing. The enhanced understanding resulting from our work might inspire new approaches to control the activation and localisation of FAK and PYK2 for therapeutic purposes.

## Methods

**Cloning**. The FAK and PYK2 DNA fragments FAK$_{764-845}$, FAK$_{776-841}$, FAK$_{804-832}$, PYK2$_{728-839}$, PYK2$_{761-839}$, and PYK2$_{789-839}$ from *Homo sapiens* were amplified using the oligonucleotide primers (IDT) summarised in Table 1. Cloning was performed using restriction digest with *Bam*HI and *Xho*I and the constructs were ligated into a pET-32a (+) modified expression vectors.

**Expression and purification**. The N-terminal 6×His fusion protein constructs FAK KFL$_{776-841}$, FAK KFL$_{804-832}$, PYK2 KFL$_{728-839}$, PYK2 KFL$_{761-839}$ FAK FERM$_{31-405}$ PYK2 FERM$_{35-367}$, cloned in the modified pET-32a (+) vector, were expressed in the *Escherichia coli* and purified as described in[68], but with the following differences: Cells were induced with 0.5 mM IPTG instead of 0.3 mM; pH 8.0 was used instead of pH 7.0 in all buffers; no imidazole was added in the wash buffer for the nickel column purification, proteins were eluted from the nickel column using 500 mM imidazole instead of 250 mM; and eluted proteins were applied to a Superdex 75 size-exclusion column (GE Healthcare) using the SEC buffer containing 20 mM HEPES pH 7.5, 200 mM NaCl, and 2 mM DTT. The constructs of FAK KFL$_{776-841}$, FAK KFL$_{804-832}$, PYK2 KFL$_{728-839}$, PYK2 KFL$_{761-839}$ with an N-terminal wild-type MBP fusion tag were cloned into pETM44 vectors and expressed and purified as described previously[43], with the difference that the MBP tag was cleaved using TEV protease at a concentration ratio of 1:100 overnight at 4 °C; that the cleaved MBP tag was removed from the protein solution by passing it through amylose beads; and that eluted protein was applied to a Superdex 75 SEC column (GE Healthcare) using the SEC buffer. The eGFP-MLCK, eGFP-PYK2 KFL$_{728-839}$ and mScarlet-CaM constructs were cloned in a SUMOstar™ vector with an N-terminal 6×His tag followed by a SUMO tag, and were purified using the protocol used for His-tagged proteins. The His and SUMO tags were cleaved by the ULP1 protease and were subjected to a Superdex 75 SEC (GE

**Table 1 List of the primers used for cloning.**

| Name | Primers | |
| --- | --- | --- |
| MBP–FAK$_{764-845}$ | Forward | CCAGGGAGCAGCCTCGATGCTGGAAGTTCTGTTCCAGGG |
| | Reverse | GCAAAGCACCGGCCTCGTCAGTCGATAGAACCACGAGACAG |
| 6×His-FAK$_{764-845}$ | Forward | ATTGGATCCCAGGCGTCTCTGCTGGACCAGACCGAC |
| | Reverse | ATTCTCGAGCTAGTCGATAGAACCACGAGACAGACG |
| MBP–FAK$_{776-841}$ | Forward | CCAGGGAGCAGCCTCGATGCTGGAAGTTCTGTTCCAGGGTCC |
| | Reverse | GCAAAGCACCGGCCTCGTCAACGAGACAGACGAACGTCCGG |
| 6×His-FAK$_{776-841}$ | Forward | ATTGGATCCCGTCCACAGGAAATCGCTATGTGG |
| | Reverse | ATTCTCGAGCTAACGAGACAGACGAACGTCCGGTTTC |
| 6×His-FAK$_{804-832}$ | Forward | ATTGGATCCCCGACCCACCTGATGGAGGAGCGT |
| | Reverse | ATTCTCGAGCTAGAAGCGTTCTTCTTTTTCCAGCCAA |
| MBP–PYK2$_{728-839}$ | Forward | CCAGGGAGCAGCCTCGACCAACCTGCTGGCGCCGAAAC |
| | Reverse | GCAAAGCACCGGCCTCGTCAAGATTTGTCGTTCATGTAAACCATCGGGTC |
| 6×His–PYK2$_{728-839}$ | Forward | ATTGGATCCACCAACCTGCTGGCGCCGAAACTC |
| | Reverse | ATTCTCGAGTCAAGATTTGTCGTTCATGTAAACCATC |
| MBP–PYK2$_{761-839}$ | Forward | CCAGGGAGCAGCCTCGAACTCTCTGCACACTCCGCCGC |
| | Reverse | GCAAAGCACCGGCCTCGTCAAGATTTGTCGTTCATGTAAACCATCGGG |
| 6×His–PYK2$_{761-839}$ | Forward | ATTGGATCCAACTCTCTGCACACTCCGCCG |
| MBP–PYK2$_{789-839}$ | Forward | CCAGGGAGCAGCCTCGTCTCGTGAAGAAGCGCAGCAGCT |
| | Reverse | GCAAAGCACCGGCCTCGTCAGTCGTTCATGTAAACCATCGGG |
| 6×His–PYK2$_{789-839}$ | Forward | ATTGGATCCTCTCGTGAAGAAGCGCAGCAGCTG |

Healthcare) using SEC buffer. PYK2 FERM–kinase and human Src constructs were cloned into modified pFastBac vector with a N-terminal 6xHis–eGFP tag followed by a HRV-3C protease cleavage site[69]. They were expressed in HighFive cells using the amplified virus from SF9 cells, which included the target genes generated by the Bac-to-Bac system. Cell lysate was incubated with 10 mL CNBr-activated Sepharose beads (GE Healthcare) coupled with anti-GFP nanobodies. The 6xHis–eGFP tag was cleaved by the HRV-3C protease at 4 °C overnight and the untagged protein was eluted by gravity from the GFP–nanobody column. Eluted proteins were further purified using Superose 6 increase 10/300 GL column (GE Healthcare) in SEC buffer. All proteins used in this study were stored at −80 °C in 20 mM HEPES pH 7.5, 150 mM NaCl, and 2 mM DTT buffer. Protein purity and molecular weights ($Mw$) were evaluated using SDS–PAGE gels, and protein identity was confirmed using mass spectroscopy. All the purification and solubility tags were proteolytically cleaved, unless mentioned otherwise. Synthetic peptides were purchased from GenScript.

**Circular dichroism (CD).** CD spectroscopic experiments were conducted in biological triplicates at 25 °C using a spectropolarimeter (JASCO) with a 0.1-mm path-length cell. FAK KFL$_{776-841}$, PYK2 KFL$_{728-839}$, and CaM constructs were dialysed into PBS buffer (10 mM Na$_2$HPO$_4$, 2 mM KH$_2$PO$_4$, 30 mM NaCl, 0.5 mM TCEP, pH 6.5 and pH 7.5, respectively). PYK2 KFL$_{728-839}$, CaM, and PYK2 KFL$_{728-839}$: CaM complex samples were measured in the presence and absence of Ca$^{2+}$. FAK KFL$_{776-841}$ was measured at 27 µM, PYK2 KFL$_{728-839}$ and CaM were measured at a concentration of 20 µM, whereas the PYK2 KFL$_{728-839}$: CaM complex was measured at 40 µM. FAK FERM$_{31-405}$ and PYK2 FERM$_{35-367}$ constructs were dialysed in 20 mM HEPES, 200 mM NaCl, 0.5 mM TCEP, pH 7.5. FAK FERM$_{31-405}$ was measured at 10 µM, and PYK2 FERM$_{35-367}$ was measured at 12 µM at 25 °C and 60 °C. CD spectra were recorded from 240 to 190 nm at an interval of 1 nm. The results were analysed using CAPITO[70] and given as CD in delta ellipticity (/M/cm).

**Analytical size-exclusion chromatography (SEC).** SEC experiments were performed using the Superdex 200 30/300 column (GE Healthcare) on an AKTA pure system using a buffer containing 20 mM HEPES, 200 mM NaCl, and 2 mM DTT at pH's 6.5 and 7.5. For CaM-binding experiments, the buffer was supplemented with 5 mM Ca$^{2+}$ or 1 mM EGTA. The protein concentration for all samples was adjusted to 5 mg/mL and samples were pre-equilibrated for 1 h in the running buffer at 25 °C. Bio-Rad Mw standards were used. The Mw of each protein of interest was determined by dividing the elution volume of the standards and unknown by the void volume, which was 7.2 mL. The outputs were plotted against the log of the Mw of the standards. The SEC-MALS data was obtained with the protein passed through Superdex 200 30/300 column (GE Healthcare) in a buffer containing 20 mM HEPES, 200 mM NaCl, and 2 mM DTT at pH's 6.5 and 7.5 through Agilent HPLC following DAWN MALS detector. Data was analysed using ASTRA software provided by the company.

**Analytical ultracentrifugation (AUC).** AUC experiments were conducted at 25 °C using an ultracentrifuge (XL-I; Beckman) equipped with a 60 Ti rotor. 200 µL of 100 µM protein sample (MBP-fusion proteins and MBP alone at 1 mg/mL) in 20 mM HEPES, 150 mM NaCl, 2 mM DTT at pH 7.5 were centrifuged for 48 h. The rotor speeds were set at 6520, 15,777, and 23,263 × $g$. The partial specific volume of the protein, solvent viscosity, and solvent density were calculated using the company-provided software "Sednterp 1.09"[71]. Data were analysed using Sedphat 10.40[72]. The monomer–dimer equilibrium model was used to fit the sedimentation profiles using rotor stretch restraints and mass conservation.

**Microscale thermophoresis (MST).** Recombinant purified 6×His–CaM, 6×His–PYK2 KFL$_{728-839}$, 6×His–PYK2 KFL$_{761-839}$, 6×His-FAK KFL$_{776-841}$, and 6×His-FAK KFL$_{804-832}$ were labelled using the NanoTemper His-tag labelling kit. Labelled proteins were kept at a concentration of 50 nM in 20 mM HEPES, 200 mM NaCl, 1 mM DTT, and 5 mM Ca$^{2+}$ or 1 mM EGTA at pH 6.5. The unlabelled protein was serially diluted in the same buffer. For dimerisation experiments in the presence of CaM, CaM was added to each sample of the dilution series to reach a final CaM concentration of 10 µM. The measurements were performed in MST premium capillaries (NanoTemper Technologies, Germany) in 50% LED power and medium MST power. Measurements were performed in biological triplicates, except for the data shown in Fig. 6b, c, where technical triplicates were used. Data were analysed using the NT Analysis software.

**Fluorescence anisotropy.** Recombinant purified 6×His–CaM, 6×His–PYK2 KFL$_{728-839}$, 6×His–PYK2 KFL$_{761-839}$, 6×His-FAK KFL$_{776-841}$, and 6×His–FAK KFL$_{804-832}$ were labelled using the NanoTemper His-tag labelling kit. Labelled proteins were kept at a concentration of 50 nM in 20 mM HEPES, 200 mM NaCl, 1 mM DTT, and 5 mM Ca$^{2+}$ or 1 mM EGTA at pH 6.5. The unlabelled protein was serially diluted in the same buffer. For dimerisation experiments in the presence of CaM, CaM was added to each sample of the dilution series to reach a final CaM concentration of 10 µM. Data were collected using spectraMax i3 (Molecular Devices) plate reader. The $K_d$ value was fitted using single binding site mode and plotted using Prism 9.0 (www.graphpad.com).

**Differential-scanning fluorimetry.** The experiment was performed as previously described[43]. Briefly, thermal stability for PYK2 KFL$_{728-839}$ and CaM was assessed in 20 mM HEPES, pH 7.5, 200 mM NaCl, 1 mM EGTA and 1 mM DTT. FAK FERM$_{31-405}$ PYK2 FERM$_{35-367}$ was measured in 20 mM HEPES, pH 7.5, 200 mM NaCl, and 1 mM DTT with temperature ranging from 25 °C to 95 °C. All proteins were used at a concentration of 10 µM, and fluorescent dye SYPRO Orange was used at a final concentration of 5X.

**Isothermal titration calorimetry (ITC).** The recombinant proteins were dialysed in 20 mM HEPES pH 7.5, 150 mM NaCl, 2 mM DTT, and 5 mM Ca$^{2+}$ or in 20 mM HEPES pH 7.5, 150 mM NaCl, 2 mM DTT, and 1 mM EGTA and further degassed before using in ITC. Titrations were performed in biological triplicates on the MicroCal ITC200 instrument at 25 °C. Synthetic peptides and proteins in the 75 µL syringe were kept at a concentration of 500 µM (PYK2 FERM, FAK FERM, PYK2 FAT and FAK FAT) or 300 µM (CaM). These molecules were injected at a rate of 0.4 µL/injection into the 0.35 mL measurement cell Proteins in the measurement cell were kept at either 50 µM (CaM, PYK2 KFL$_{761-839}$, and FAK KFL$_{776-841}$) or 30 µM (PYK2 KFL$_{728-839}$). Control experiments were performed by injecting the same ligands into buffer. The Origin 7.0 software was used for data analysis.

**Nuclear magnetic resonance (NMR).** Cells were grown with $^{15}N$-NH$_4$Cl and $^{13}C_6H_{12}O_6$ dissolved in M9 minimal medium solution, induced at OD$_{600nm}$ = 0.7 with 500 μM IPTG and harvested after overnight incubation at 18 °C. Protein samples were purified, and NMR samples were prepared by dissolving the $^{15}N$-labelled protein in a 5% D$_2$O/95% H$_2$O solution in the conditions and concentrations described in Supplementary Table 5. NMR data were collected at 25 °C on the Bruker Avance III 950 MHz spectrometer equipped with a 5-mm z-gradient TCI CryoProbe. NMR samples contained ~0.25 mM $^{15}N$-labelled or $^{13}C$-$^{15}N$-labelled protein dissolved in 20 mM HEPES buffer (pH 6.5), 150 mM NaCl, and 1 mM TCEP with 5% D$_2$O for the lock. $^1H$ chemical shifts were directly referenced to the methyl resonance of DSS, whereas $^{15}N$ and $^{13}C$ chemical shifts were referenced indirectly to the absolute $^{15}N/^1H$ and $^{13}C/^1H$ frequency ratios, respectively. All NMR spectra were processed using TopSpin (Bruker) and analysed using CINDY (http://abcis.cbs.cnrs.fr/CINDY/; Padilla, CBS). Backbone and side-chain resonance assignments were performed using standard 3D NMR experiments (HNCA, HNCACB, HN(CO)CA, CBCA(CO)NH, HNCO, HN(CA)CO, and [$^1H$$^{15}N$] NOESY–HSQC). Changes in chemical shifts for analysing CSP data for $^1H$ and $^{15}N$ were measured in ppm ($\Delta H$ and $\Delta N$). The $^{15}N$ shift changes were multiplied by a scaling factor of 0.2 and then the total change in CSP was calculated as follows: $(CSPi) = \triangle HN = \sqrt{\frac{1}{2}\left[\delta_H^2 + \left(\alpha.\delta_N^2\right)\right]}$ (1)[73]. IQR values were calculated by the equation $IQR = Q_1 - Q_3$ (2), where $Q_1$ is the first quartile, and $Q_3$ is the third quartile. The median values were calculated and data was plotted using Prism 9.0 (www.graphpad.com). To plot the 2D [$^1H$-$^{15}N$] HSQC ratios of PYK2 KFL$_{728–839}$ peak intensities at 10 μM divided by those at 100 μM, the intensity values were retrieved by exporting data height with signal-to-noise values using NMRFAM-SPARKY[74]. The intensity and signal-to-noise values for both concentrations of PYK2 KFL$_{728–839}$ were normalised according to the number of scans[75,76].

NOE cross-peaks identified on the NOESY spectra (mixing time 150 ms) were assigned via automated NMR structure calculations with CYANA 3[77]. Backbone $\varphi/\psi$ and side-chain $\chi1$ torsion angles constraints were obtained from a database search procedure on the basis of backbone ($^{15}N$, H$_N$, $^{13}C'$, $^{13}C\alpha$, H$\alpha$) and $^{13}C\beta$ chemical shifts using TALOS-N[78]. A total of 100 3D structures were generated using the torsion angle dynamics protocol of CYANA 3 from 374 NOEs, 18 hydrogen bonds, and 81 angular restraints (26 $\varphi$, 26 $\psi$, and 29 $\chi1$) for PYK2 KFL$_{728–839}$. The 20 best structures (based on the final target penalty function values) were minimised using CNS 1.2 according to the RECOORD procedure[79].

**Visible immunoprecipitation (VIP).** Purified recombinant eGFP-tagged PYK2 KFL$_{728–839}$ or the eGFP-tagged human myosin light chain kinase (MLCK) fragment RRKWQKTGHAVRAIGRLSS (taken from PDB 2K0F) were mixed with anti-GFP nanobody beads. For binding assays, mScarlet-CaM was mixed with the beads and incubated at 4 °C for 2 h in 20 mM HEPES, 150 mM NaCl, 1 mM DTT, 10 mM Ca$^{2+}$ or 10 mM EGTA. The beads were thoroughly washed with wash buffer (20 mM HEPES, 150 mM NaCl, 1 mM DTT, 10 mM Ca$^{2+}$ or 20 mM HEPES, 150 mM NaCl, 1 mM DTT, 10 mM EGTA) and then visualised under the red and green filters adapted to the excitation wavelengths of eGFP at 488 nm and mScarlet at 569 nm. The experiments were performed in the presence of 10 mM Ca$^{2+}$ or of 10 mM EGTA.

**Cell culture.** For pulldown assay, HEK293T cells were seeded in 6-cm dishes, at a density of 400,000 cells per dish, and maintained in DMEM cell culture medium (Gibco), supplemented with 10% fetal bovine serum (Gibco) and 0.2% penicillin/streptomycin (Gibco), in a humidified atmosphere at 5% CO$_2$ and 37 °C. Cells were transiently transfected 24 h after seeding with 8 μL Lipofectamine 2000 (Invitrogen) per 4 μg plasmid DNA as previously described[37,38]. Forty-eight h after transfection, cells were harvested with 300 μL ice-cold lysis buffer containing 50 mM Tris-base pH 8.0, 100 mM NaCl, 1 mM EGTA, 0.1%, v/v, Triton X-100, Complete Mini EDTA-free (Sigma) and PhosSTOP (Sigma). Cell membranes were disrupted by freeze-thaw, followed by pipetting, and protein extracts were cleared by centrifugation at 11,000 x g at 4 °C for 15 min. Protein concentration was quantified using the Micro BCA Protein Assay Kit (Thermo Scientific). For treatment with CaM antagonist, PC12 cells were grown on type I collagen-coated dishes (BD Biosciences) in RPMI medium (Gibco) containing 10% horse serum, 5% fetal calf serum. Transfection was performed as described above with GFP-PYK2 WT when cells were at about 70% confluency. Forty-eight hours after transfection, cells were treated with 50 μM calmidazolium (Cz) for 30 min, followed by depolarisation by isosmotic replacement of 40 mM NaCl by 40 mM KCl in the extracellular medium for 3 min, as previously described[37]. For this procedure, half the medium was replaced by a solution containing 1 mM MgCl$_2$, 2 mM CaCl$_2$, 25 mM HEPES, and either 135 mM NaCl, (control solution), or 55 mM NaCl and 80 mM KCl (high [K$^+$]). PC12 cells were lysed on ice, in RIPA buffer, followed by quantification, electrophoresis and immunoblot.

**Pulldown assay and immunoblot.** CaM-Sepharose 4B beads (Sigma) were left to equilibrate at room temperature (RT) for 1 h, mixed thoroughly and 50 μL beads slurry was dispensed in 2-mL centrifuge tubes. Beads were washed with pulldown buffer (50 mM Tris-base pH 8, 100 mM NaCl, Complete Mini EDTA-free, PhosSTOP) supplemented with either 2 mM CaCl$_2$ or 10 mM EGTA. Equal amounts of

protein were added to the tubes containing the CaM beads (200 μg from each cell transfection; 100 μg from each recombinant purified protein), volume was adjusted to 1 mL with respective pulldown buffers, supplemented with CaCl$_2$ or EGTA, and the suspensions were incubated for 2.5 h at 4 °C with light agitation. The beads with bound proteins were collected by centrifugation at 18,000 x g for 10 min, followed by three washes with a buffer of the same composition as used in each incubation, and followed by incubation with 50 μL of opposite buffer and 10 μL 100 mg/mL SDS (Sigma) for 30 min at RT with light agitation. In parallel, 1% of protein from cell transfection or of recombinant purified protein were collected into clean centrifuge tubes. After the last incubation, 4x Laemmli buffer was added to input and beads fractions and samples were heated at 95 °C for 10 min. Input and bead (Ca$^{2+}$ or EGTA) fractions from cell transfections were subjected to electrophoresis in denaturing conditions (SDS–PAGE) and transferred to nitrocellulose membranes for immunoblotting. Membranes were saturated in 25 g/L BSA in TBS with 0.1% Tween-20 (TBS-T), followed by washes with TBS-T, overnight incubation at 4 °C with primary antibody chicken anti-GFP (#A10262, Invitrogen) at 1:1000, washes with TBS-T, 1 h incubation at RT with primary antibody mouse anti-tubulin (#T9026, Sigma) at 1:10000, washes with TBS-T, 2 h incubation at RT with secondary antibodies anti-chicken 800 nm (#926-32218, LI-COR) and anti-mouse 680 nm (#926-68072, LI-COR), both at 1:5000, washes with TBS-T, and fluorescence acquisition with Odyssey CLx (LI-COR). Protein extracts from Cz treatment and depolarisation experiments were submitted to electrophoresis, but were immunoblotted with primary antibody rabbit anti-PYK2 (#P3902, Sigma) or primary antibody Rabbit anti-pY402 PYK2 (#44-618G, Invitrogen) at 1:1000, followed by secondary antibodies anti-rabbit 800 nm (#926-32213, LI-COR) at 1:5000. Protein band quantification was performed using Image Studio Lite 5.2 (LI-COR). For quantification of experiments of binding of GFP-PYK2 (WT or deleted forms) to CaM beads in the presence of Ca$^{2+}$ or EGTA, the amount of GFP-immunoreactivity associated with the beads was divided by the amount of GFP-immunoreactivity in the input for correction of the variable amounts of fusion protein depending on the constructs. For the experiments in which phosphorylation of PYK2 in response to depolarisation was studied in the absence or presence of Cz, pY402 immunoreactivity was divided by total PYK2 immunoreactivity. Input and bead (Ca$^{2+}$ and EGTA) fractions from recombinant purified protein were submitted to electrophoresis, stained using InVision His-Tag In-Gel stain and observed using BioRad Gel dock imager at 260 nm.

**Bioinformatics.** The sequence alignment of the 78 sequences between FAK and PYK2 was performed using the online server MAFFT https://mafft.cbrc.jp/alignment/software/. The alignment result was visualised using Guidance 2.0 http://guidance.tau.ac.il/ver2/. Secondary structure predictions were made with PHD[80], PSIPRED[81], JPred4[82], and AlphaFold2[83], while coiled-coil prediction was made using COILS[84] and Waggawaga webserver for NCOILS[85]. CaM-binding motifs were taken from cam.umassmed.edu[86].

**Statistics and reproducibility.** Statistical data analysis was performed using GraphPad Prism 9.0 and Microsoft Excel 16.62. The statistical data values are given as mean ± standard error (SD) of the mean, where 'n' is the number of replicates indicated in applicable figure legends. Curve fitting for binding experiments was performed using a one-site total non-linear fit.

**Reporting summary.** Further information on research design is available in the Nature Research Reporting Summary linked to this article.

## Data availability

NMR assignments of PYK2 KFL$_{728–839}$ are deposited in the Biological Magnetic Resonance Data Bank (BMRB accession code: 50961). The source data underlying graphs, plots, and charts in the manuscript are presented in Supplementary Data.

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

## Acknowledgements

We acknowledge SOLEIL for providing synchrotron radiation facilities (proposals nr. 20181104 and 20180576). We would also like to thank J. Perez and A. Thureau for assistance in using the beamline SWING. We thank the KAUST Bioscience, and Imaging and Characterisation core labs for their assistance. We thank T.M.D. Besong and O. Bakr from the Functional Nanomaterials Lab at KAUST for their help with the AUC measurement and analysis. We thank D. Renn and M. Rueping (KAUST Catalysis Center) for the access to the CD instrument. We also thank U.S. Hameed for discussions on the manuscript and R. Naser for help with NMR data. This publication is based on the work supported by the King Abdullah University of Science and Technology (KAUST) Office of Sponsored Research (OSR) under the Award Number URF/1/2602-01-01. The work in JAG's lab was supported by grants from *Agence Nationale de la Recherche* (ANR-19-CE16-0020) and *Fondation pour la Recherche Médicale* (FRM, EQU201903007844). P.B. acknowledges support from the French Infrastructure for Integrated Structural Biology (FRISBI) ANR-10-INSB-05.

## Author contributions

Protein cloning, expression and purification: A.A.M. and S.H. Biochemical and biophysical analysis: A.A.M., S.H., P.Y. NMR analysis: A.A.M., P.B., M.J., Ł.J. SAXS analysis: A.A.M., S.T.A. Cell biology: T.M., C.F., G.K., J.A.G. Project design and supervision: S.T.A., J.A.G., P.B., M.J., Ł.J. Manuscript writing: A.A.M., S.T.A., T.M., J.A.G., Ł.J. All authors read and approved the manuscript.

## Competing interests

The authors declare no competing interests.
