## [Peer Review File · Communications Biology]

Reviewers' comments:

Reviewer #1 (Remarks to the Author):

The authors seek to clarify mechanisms by which the PYK2 non-receptor tyrosine kinase responds to calcium flux. As the introduction nicely summarizes, PYK2 is a product of FAK gene duplication. Both FAK and PYK2 share an overall domain organization, and significant sequence homology. However, PYK2 appears to have adopted calcium sensitivity as part of its activation mechanism, a phenomenon noted in 1995 (Ref 11). Nevertheless, although several mechanistic models have been proposed (usually invoking calmodulin, CaM, as an intermediary calcium-sensitive signaling effector), previously reported data is contradictory, and the mechanisms of the unique calcium sensitivity of PYK2 remain unresolved. This is a broadly important question for two reasons: 1) FAK and PYK2 are implicated in cancer metastasis and neuronal signaling, and 2) the unique PYK2 functionality is an excellent example of genetic "duplication and diversification."

The authors explore CaM-mediated calcium sensitivity in PYK2 by shifting focus to the largely unstructured linker between kinase and FAT domains (KFL). Using a battery of pull-downs and biophysical approaches, they reveal that the KFL can bind CaM directly. The KFL-CaM interaction that does not require calcium, but calcium strengthens the interaction. Notably, the corresponding FAK KFL does not bind CaM. CaM pull-downs from cell lysate complicate the story, showing varying degrees of interaction with PYK2 truncations sampling FERM and kinase domains, with or without the KFL.

Focusing on the KFL, the authors find that the largely unstructured region migrates as a loose dimer via SEC-MALS and AUC. NMR experiments were also employed to map dimerization and CaM-binding regions. Collectively, the NMR experiments suggest that CaM interacts with the PYK2 KFL via general electrostatics, and calcium rearranges the interaction to engage additional residues. The authors propose a very novel "fuzzy" interaction that can tune KFL-mediated dimerization in response to calcium flux.

Overall, this is a clearly written study that proposes a new role for the PYK2 KFL in kinase calcium regulation. The data and model proposed herein nicely circumvent contradictions in previously reported calcium regulation hypotheses. In my view, the most significant results here are the demonstrations that the CaM-binding of the KFL is unique to PYK2, differentiated from FAK. The balance between KFL-mediated dimerization, CaM-binding, and the other proposed inter-domain interactions involved in PYK2 regulation remains speculative. Nevertheless, the Discussion section provides a welcome and thought-provoking model integrating these new results and previously proposed mechanisms of PYK2 regulation. This is an important contribution to the field of FAK/PYK2 signaling. I expect this example of functional specialization in paralogs will be broadly interesting to signaling enthusiasts. Likewise, the curious "fuzzy" interactions of the disordered KFL serves as another example in the growing area of macromolecular recognition in intrinsically-disorder proteins.

I have a few minor points that could clarify the manuscript:

1. My primary point of confusion is about the protein constructs used in these studies. The Methods section does not clearly distinguish which constructs were expressed with which purification/solubility tags. Further, it is not clear whether these tags have been removed before the experiments. Two additions would resolve this.

1a. The Methods section relies on two references to provide general protein preparation details. However, these references don't resolve which constructs were purified with which tags. Furthermore, the His-tag reference seems to include several purification methods. And the MBP reference #43 is a (useful) paper warning about MBP-derived dimerization artifacts. Can the authors provide more details in the protein purification Methods section? Also, were the MBP constructs modified to avoid the MBP-dimer artifact revealed in Ref. #43?

1b. It does seem likely that the inclusion of tags is explicitly stated in the figures (for example, Supplementary Figure 6a). It would be helpful to clarify that this convention is consistent

throughout the manuscript with a statement in the Methods section (for example, something like "purification and solubility tags were proteolytically cleaved unless noted otherwise.")

2. The authors propose that FERM-containing PYK2 fragments retained on CaM-beads (from lysate) may be a result of indirect interactions. Is it possible that endogenous PYK2 is contributing to this indirect interaction via FERM-FERM or FAT-FERM interactions? I am unsure whether this cell line expresses endogenous PYK2.

3. Fig. 2C shows PYK2 KFL(728-839) eluting with an apparent MW of 37 kDa. Is this the dimer or a MW shift due to a purification tag? If it's the dimer, this would warrant a mention in the main Results text.

4. On page 5, a reference is made to insoluble FAK KFL(746-845). Is this a typo? Fig. 2a lists a FAK FKL(764-845).

5. Supplementary Figure 6a seems to have a typo in the protein construct labeling. Both red and blue traces are labeled as the same construct.

6. On page 11, The last Results sentence refers to Fig. 6c, but that is supposed to be 6d, right? Likewise, the following sentence seems to be referring to Fig. 6c rather than S9a: "However, monomeric PYK2 KFL728-839 bound to the PYK2 FERM-kinase fragment with micromolar affinity (Supplementary Fig. 9a)"

7. To better assess analytical SEC results (especially in Fig. 2c), more experimental details would be useful. In addition to the aforementioned clarifications on whether constructs are tagged or not, the Methods section would benefit from some details about protein concentrations and binding pre-equilibration times/conditions.

Reviewer #2 (Remarks to the Author):

The work by Momin and coworkers use a multidisciplinary approach to study how calmodulin interacts with a disordered region in the tyrosine kinase PYK2 and from their results they suggest a model that includes dimerization of PYK2 and where calmodulin can activate the kinase. The work combines cell biology and a number of biophysical methods including NMR, SAXS, ITC, MST and FP to investigate the different complexes formed. The complexes studied are interesting, the manuscript is well written and the data seem to have been conducted in a meticulous way. However, a number of controls are missing, and the study critically fails to address the dynamics of the complexes as explained below. Also, the authors seem to have missed a number of essential studies in the literature. So, while the study may be of potential interest, several additional experiments are needed to bring the work to completion.

1. Mapping chemical shift perturbations in complexes involving IDPs as a way to map interactions is problematic: first, because a chemical shift change may occur not because of binding, but because of ensemble redistribution. Second, due to the dynamics of the system, the bound state may not be visible in the NMR spectra as chemical exchange on an intermediate time scale may lead to disappearance of the signal. Thus, the shift mapping – also in the case of the monomer-dimer equilibrium – should be accompanied at least by plots of intensity ratios (as in Figure 5g) and of R2 measurements, but full relaxation data is preferable. In the cases where the peaks are all disappearing (as in Figure 5g), measurement of the changes in R2 at lower saturation ratios can be done to map the binding sites with more confidence (as done in e.g. Delaforge JACS 2018). As it stands, I am not convinced that the authors have identified the correct binding interfaces from the CSPs – the scattering of the perturbation across the sequence and the many missing signals certainly suggest that the bound state is not identifiable in the NMR spectra (no signal at the 1:2 ratio in Figure 5g). Changing the salt concentration or temperature may change the exchange regime

2. Similar goes for the monomer-dimer equilibrium. There are not any zooms on peaks that move

during the dilution so it is not possible to assess the data. Please include this to show saturation. Here, again, intensity ratios and R2 values should be recorded so the correct dimerization interface is mapped. There are several data in the literature on disordered dimers (see e.g. Danielsson et al., *Biochemistry* 2008), so this is not a first observation made.

3. Figure 1 provides the foundation for the study, however the Western Blot analyses are unclear, especially with respect to the quantification performed in Figure 1C. The representative image does not appear to match the quantified data, i.e., comparing the KFL700-841 band with PYK2 wt1-1009. It is also necessary to show the loading control for each blot. Western Blot analyses could be described better in the methods with respect to normalization to control bands.

4. The authors describe the KFL as extremely/exceptionally disordered. What are the underlying data that support the extreme/the exceptional disorder? It is disordered with a rather folded helix populated to at least 50% (or more). Why is it not just disordered? I think this is a more appropriate term. Also, why is this unusually flexible? Again, it is likely just disordered. There are others vaguely formulated terms as "peculiar features" (p.12 – what are they? And why are they peculiar?); "CaM caressing" – what does this mean? There are reports available on disordered complexes with disordered regions binding to e.g. ERK2 and to virus proteins (Milles, *Cell*; Hendus-Altenburger, *BMC biology*). How is the complex reported here different to those?

5. What is the new paradigm in CaM binding and why is this unusual/non-conventional? This is not clear from the text and because the bound state is likely invisible, details are missing that makes the conclusions hard to reach. In recent years, several studies have highlighted a huge diversity in CaM binding distinct from the helix-binding motifs, and there are reports on the existence of several different states interconverting even with different stoichiometries, including also binding to the calcium free state (see e.g. Lee...Ghose, *Protein Science* 2019; Nunomoura et al., *BJ*, 2014; Sjogaard-Frich, Prestel et al., *eLife*, 2020), and oligomerization involving CaM (Barros et al., *IJMS* 2019; Li...Sacks, *JBC* 2017). None of these reports are cited. A broader view on CaM binding is already emerging in the literature and should be referred to in the text.

6. In line with these issues, the claimed fuzzy interaction has not been supported by data. Relaxation measurements, diffusion, relaxation dispersion or molecular simulations, or smFRET data is needed to conclude this. The issue is if this interaction is just dynamic (fast on-off rates) or if there are multiple conformations of the bound state, that all contribute to the binding (which then suggest the interaction to be fuzzy). Please support these claims with data. If no folding occurs on binding, how do you explain the favorable enthalpy change?

7. CD analyses have been provided for KFL domains alone, but it would be beneficial to add CD analysis of the PYK2 KFL domain interacting with CaM to demonstrate that KFL in the interaction remains disordered. This is particularly important as the bound state is inaccessible by NMR.

8. Figure 2B is highly oversaturated and does not show co-localization. Please fix and provide an overlay to make this clearer.

9. Is the dimer broken or intact in the complex? I think this is relevant to address as a dimer here suggest similar mechanism as seen e.g in CaM binding to the estrogen receptor and in other membrane proteins.

10. The dispute regarding the role of CaM in regulation of PYK2 is not explained well enough that the reader can understand and appreciate the potential clarification this work may contribute with, please include.

11. Does KFL bind calcium?

12. Is the binding of KFL to calcium-free CaM biologically relevant? No binding to CaM is seen in the presence of EGTA in the full-length protein.

13. The lack of binding of the FERM domain could be due to the fact that the recombinant protein is not correctly folded. What data can be included to show this? CD, SEC.

14. For all binding experiments, the number of replicates (n) should be stated in the figure legend and the errors listed explained (are they errors of the fit, propagated errors, or standard errors of the mean, etc)

15. It is possible – and interesting - that there are flanking region effect as context is emerging as playing key roles in affinity for IDPs. This should be substantiated once the binding regions have been more firmly established.

Minors.

p. 2: several linear interaction motifs – Do you mean short linear motifs?

p. 7. Primary sequence -> primary structure or just sequence

Figure 4: used 250 mM

Where did the author purchase their peptides?

No DTT (or TCEP) was added to the CD analyses, why not?

Figures could be easier to understand: e.g. providing a color legend on figure 5F, 5H; better demarcation of calcium in figure 6D (left).

Reviewer #3 (Remarks to the Author):

This review is for the submitted manuscript "PYK2 senses calcium through a highly disordered dimerization and calmodulin- binding element" written by Dr. Stefan Arold, et al. This manuscript is a well written, scientifically important study and I enthusiastically support publication.

The major finding of the paper is the identification of a novel CaM binding element Specifically, "PYK2 KFL is highly disordered and engages CaM through an ensemble of transient binding events." The authors also found that calcium increases this association by promoting structural changes in CaM.

These claims are supported by strong experimental evidence including a series of high quality NMR experiments.

One potential addition that could strengthen the manuscript would be to add NMR-based dynamics analysis, as changes in dynamics is suggested in the text but not supported experimentally by NMR.

Regardless, it is my belief that this work would be of interest to the field and provides an interesting perspective regarding the interaction of CaM with target proteins.

Detailed response to reviewers

Reviewer #1 (Remarks to the Author):

The authors seek to clarify mechanisms by which the PYK2 non-receptor tyrosine kinase responds to calcium flux. As the introduction nicely summarizes, PYK2 is a product of FAK gene duplication. Both FAK and PYK2 share an overall domain organization, and significant sequence homology. However, PYK2 appears to have adopted calcium sensitivity as part of its activation mechanism, a phenomenon noted in 1995 (Ref 11). Nevertheless, although several mechanistic models have been proposed (usually invoking calmodulin, CaM, as an intermediary calcium-sensitive signaling effector), previously reported data is contradictory, and the mechanisms of the unique calcium sensitivity of PYK2 remain unresolved. This is a broadly important question for two reasons: 1) FAK and PYK2 are implicated in cancer metastasis and neuronal signaling, and 2) the unique PYK2 functionality is an excellent example of genetic “duplication and diversification.”

The authors explore CaM-mediated calcium sensitivity in PYK2 by shifting focus to the largely unstructured linker between kinase and FAT domains (KFL). Using a battery of pull-downs and biophysical approaches, they reveal that the KFL can bind CaM directly. The KFL-CaM interaction that does not require calcium, but calcium strengthens the interaction. Notably, the corresponding FAK KFL does not bind CaM. CaM pull-downs from cell lysate complicate the story, showing varying degrees of interaction with PYK2 truncations sampling FERM and kinase domains, with or without the KFL.

Focusing on the KFL, the authors find that the largely unstructured region migrates as a loose dimer via SEC-MALS and AUC. NMR experiments were also employed to map dimerization and CaM-binding regions. Collectively, the NMR experiments suggest that CaM interacts with the PYK2 KFL via general electrostatics, and calcium rearranges the interaction to engage additional residues. The authors propose a very novel “fuzzy” interaction that can tune KFL-mediated dimerization in response to calcium flux.

Overall, this is a clearly written study that proposes a new role for the PYK2 KFL in kinase calcium regulation. The data and model proposed herein nicely circumvent contradictions in previously reported calcium regulation hypotheses. In my view, the most significant results here are the demonstrations that the CaM-binding of the KFL is unique to PYK2, differentiated from FAK. The balance between KFL-mediated dimerization, CaM-binding, and the other proposed inter-domain interactions involved in PYK2 regulation remains speculative. Nevertheless, the Discussion section provides a welcome and thought-provoking model integrating these new results and previously proposed mechanisms of PYK2 regulation. This is an important contribution to the field of FAK/PYK2 signaling. I expect this example of functional specialization in paralogs will be broadly interesting to signaling enthusiasts. Likewise, the curious “fuzzy” interactions of the disordered KFL serves as another example in the growing area of macromolecular recognition in intrinsically-disorder proteins.

I have a few minor points that could clarify the manuscript:

1. My primary point of confusion is about the protein constructs used in these studies. The Methods section does not clearly distinguish which constructs were expressed with which purification/solubility tags. Further, it is not clear whether these tags have been removed before the experiments. Two additions would resolve this.

- 1a. The Methods section relies on two references to provide general protein preparation details. However, these references don't resolve which constructs were purified with which tags. Furthermore, the His-tag

reference seems to include several purification methods. And the MBP reference #43 is a (useful) paper warning about MBP-derived dimerization artifacts. Can the authors provide more details in the protein purification Methods section? Also, were the MBP constructs modified to avoid the MBP-dimer artifact revealed in Ref. #43?

Thank you for pointing this out. We have now updated the methods section to explicitly associate the tag and vector used with the name of each construct.

The MBP tag used for some constructs in this study is the wild-type MBP. As mentioned in our report Momin et al. Sci Rep 2020, WT MBP does not show the propensity to dimerize that we detected in the engineered MBP. We have now clearly stated in the Methods that we used wild-type MBP.

1b. It does seem likely that the inclusion of tags is explicitly stated in the figures (for example, Supplementary Figure 6a). It would be helpful to clarify that this convention is consistent throughout the manuscript with a statement in the Methods section (for example, something like “purification and solubility tags were proteolytically cleaved unless noted otherwise.”)

This is an important point. We mention the tags when these were included in a study, for example, as MBP-PYK2 KFL. However, we agree that it is beneficial to add “All the purification and solubility tags were proteolytically cleaved, unless mentioned otherwise.” in the method section and have done so.

2. The authors propose that FERM-containing PYK2 fragments retained on CaM-beads (from lysate) may be a result of indirect interactions. Is it possible that endogenous PYK2 is contributing to this indirect interaction via FERM-FERM or FAT-FERM interactions? I am unsure whether this cell line expresses endogenous PYK2.

There was no detectable endogenous PYK2 in the HEK293 cells used for these experiments (see Fig. 1b which shows the input fraction, which corresponds to the total solubilized proteins from the HEK293 culture). To clarify this point we have added the following sentence: “These cells did not express detectable amounts of endogenous Pyk2 (Fig. 1b).” at the beginning of the Results section (second sentence).

3. Fig. 2C shows PYK2 KFL(728-839) eluting with an apparent MW of 37 kDa. Is this the dimer or a MW shift due to a purification tag? If it's the dimer, this would warrant a mention in the main Results text.

The analytical SEC in Fig2C was performed without tag on the protein construct. Hence, the molecular weight which we deduce corresponds to self-association, and not the added Mw from a tag. The absence of a tag should be clear now, as we have included the specific statement in the Methods section (“All the purification and solubility tags were proteolytically cleaved, unless mentioned otherwise.”).

Following this reviewer's suggestion we have also added the following statement to the Results section: “In our analytical SEC analysis (Fig. 2c, middle panel) PYK2 KFL₇₂₈₋₈₃₉ eluted with a molecular weight markedly greater than expected for a monomer, suggesting self-association.” (first sentence of section *PYK2 KFL forms extremely disordered dimers*).

4. On page 5, a reference is made to insoluble FAK KFL(746-845). Is this a typo? Fig. 2a lists a FAK FKL(764-845).

Indeed, it was a typo; it was meant to be FAK KFL₇₆₄₋₈₄₅. We have rectified it in the text.

5. Supplementary Figure 6a seems to have a typo in the protein construct labeling. Both red and blue traces are labeled as the same construct.

We have rectified the figure, and it is labeled with the correct construct names in the revised version.

6. On page 11, The last Results sentence refers to Fig. 6c, but that is supposed to be 6d, right? Likewise, the following sentence seems to be referring to Fig. 6c rather than S9a: "However, monomeric PYK2 KFL728-839 bound to the PYK2 FERM–kinase fragment with micromolar affinity (Supplementary Fig. 9a)"

Thank you - we have rectified the mistakes and have appropriately mentioned the figure numbers in the text.

7. To better assess analytical SEC results (especially in Fig. 2c), more experimental details would be useful. In addition to the aforementioned clarifications on whether constructs are tagged or not, the Methods section would benefit from some details about protein concentrations and binding pre-equilibration times/conditions.

The purification and solubility tags were proteolytically cleaved, unless mentioned otherwise, and we have included this sentence in the methods section for clarity as suggested (please see our replies above). But we agree that more experimental details would be useful. Hence, we have additionally included the concentrations used for all proteins as well as pre-equilibration times and conditions: "The protein concentration for all samples was adjusted to 5 mg/ml and samples were pre-equilibrated for 1 hour in the running buffer at 25°C."

Reviewer #2 (Remarks to the Author):

The work by Momin and coworkers use a multidisciplinary approach to study how calmodulin interacts with a disordered region in the tyrosine kinase PYK2 and from their results they suggest a model that includes dimerization of PYK2 and where calmodulin can activate the kinase. The work combines cell biology and a number of biophysical methods including NMR, SAXS, ITC, MST and FP to investigate the different complexes formed. The complexes studied are interesting, the manuscript is well written and the data seem to have been conducted in a meticulous way. However, a number of controls are missing, and the study critically fails to address the dynamics of the complexes as explained below. Also, the authors seem to have missed a number of essential studies in the literature. So, while the study may be of potential interest, several additional experiments are needed to bring the work to completion.

1. Mapping chemical shift perturbations in complexes involving IDPs as a way to map interactions is problematic: first, because a chemical shift change may occur not because of binding, but because of ensemble redistribution. Second, due to the dynamics of the system, the bound state may not be visible in the NMR spectra as chemical exchange on an intermediate time scale may lead to disappearance of the signal. Thus, the shift mapping – also in the case of the monomer-dimer equilibrium – should be accompanied at least by plots of intensity ratios (as in Figure 5g) and of R2 measurements, but full relaxation data is preferable.

We agree that these are important points, and we addressed them in several ways (as detailed below).

Firstly, to address the possibility for intermediate exchange, we recorded the same spectrum at two magnetic fields (700 and 950 MHz) to see if we observe additional signals when moving to the higher field (which would move the shifts towards slow exchange). Such an effect would support the hypothesis of intermediate exchange, with the bound form signals being broadened beyond detection. However, we do not observe new signals appearing at 950 MHz, or significant narrowing of the signals at the higher field. Based on the dissociation K_d from MST and fluorescence anisotropy (0.8 μM), the bound state is populated to 94 % at the

100 μM concentration used. Therefore, we can state that the vast majority of the observed signals belong to the bound form and that this form is highly flexible.

We have now added the comparison between 700 and 950 MHz as **Supplementary Figure 10**. And we have added the following text in the Results section *PYK2 KFL forms disordered dimers*: “To address the possibility that an intermediate exchange regime concealed the resonances of the bound state at the 700 MHz frequency used, we also recorded the 100 μM [^{15}N]-PYK2 KFL728-839 sample at 950 MHz. We neither observed new signals appearing, nor a significant narrowing of signals at 950 MHz, supporting that our analysis was not affected by a significant contribution of intermediate exchange (Supplementary Fig. 10).”

Supplementary Figure 10. [^1H , ^{15}N] HSQC overlay for [^{13}C , ^{15}N] PYK2 KFL₇₂₈₋₈₃₉ at 100 μM concentration measured at 950 MHz (blue) and 700 MHz (red).

We did consider determining the transverse relaxation rate. This rate is known to be sensitive to the tumbling time, and hence to the molecular weight/shape of the solute species. However, the strength of the association is so high ($K_d=0.8 \mu\text{M}$) that even if we dilute the KFL in the sample to 10 μM (we cannot use less due to the sensitivity loss), the sample will still contain less than 20% of the monomeric state contributing to the R2. However, changing the protein concentration 5 times will significantly decrease the viscosity, likely masking the transverse relaxation rate differences between monomeric and dimeric mixture (approximately 94 % dimer at 100 μM and 82 % of the dimer at 10 μM). Moreover, the error of the measurement will increase with lower protein concentration, making our results even less conclusive. We agree that others have used this type of measurement successfully, but for a system with a much higher dimerisation K_d (e.g. 100 fold higher in Danielsson et al., Biochemistry 2008). Therefore, although this type of measurement is in principle very powerful, it will have only very limited value for testing our conclusion. Finally, we note that the existence of the dimer is further verified by many other measurements, such as MST, fluorescence anisotropy, AUC, SEC-MALS and SAXS (Figures 2,3 and Supp Figures 2,5,6,7).

We did follow the reviewer's suggestion to add the plot of the intensity change observed upon diluting the dimeric KFL sample from 100 μM to 10 μM . We have done so in the revised version. As noted above (and now in the manuscript) due to experimental constraints we can only compare state changes from 94 % dimer to 82% dimer, expecting the signal to be small in both CSPs and intensity measurements. When comparing the intensity change with the CSPs occurring as a result of this dilution, we observed the same features, namely that no single region experiences strong and consistent shifts. Rather, there are a few regions that the data suggest participate in the interaction. Although there is some level of agreement on these regions between CSP and intensity analyses, the data support that the dimerisation is fuzzy and not very specific, supporting our conclusions. Accordingly, we have now added the intensity plot to figure 4B:

Figure 4 (b) (Top) CSP occurring as a result of diluting [^{13}C , ^{15}N] PYK2 KFL₇₂₈₋₈₃₉ from 100 μM (8 scans) to 10 μM (128 scans), monitored by 2D [^1H - ^{15}N] HSQC. The orange horizontal line indicates the median threshold for minor shifts and the horizontal green line and shows the interquartile range (IQR) threshold for major shifts. **(Bottom)** Plot of 2D [^1H - ^{15}N] HSQC intensity ratios of [^{13}C , ^{15}N] PYK2 KFL₇₂₈₋₈₃₉. The ratios were calculated as 10 μM (128 scans) divided by 100 μM (8 scans). The intensity value at 100 μM was taken as a reference and used to normalise the intensity value at 10 μM to compensate for an overall loss of intensity upon dilution. Red dotted line indicates the median. Grey shaded zones indicate residue regions where significant CSPs correspond to regions of lower relative intensity

We have also added the following to our Results section (page 8): “The regions of significant CSP changes broadly correlated with an analysis of the peak intensity change as a result of sample dilution (Fig. 4b, bottom). Both methods revealed that the dilution affected many regions relatively weakly, and that these regions were scattered across the length of the molecule. We concluded that PYK2 KFL₇₂₈₋₈₃₉ dimerization is a fuzzy and rather unspecific process.”

We also agree that we needed to state more clearly that our data only allow a relatively tentative assignment of the regions identified as contributing in dimerisation. We have therefore added the following to the Discussion section: “The sub-micromolar dimerization K_d precluded a full relaxation analysis, and hence a confident assignment of the residues involved in dimerization. However, CSPs and intensity changes showed that the dimerization results from many poorly specific associations that are dispersed along the whole PYK2 KFL₇₂₈₋₈₃₉ region.”

In the cases where the peaks are all disappearing (as in Figure 5g), measurement of the changes in R2 at lower saturation ratios can be done to map the binding sites with more confidence (as done in e.g. Delaforge JACS 2018). As it stands, I am not convinced that the authors have identified the correct binding interfaces from the CSPs – the scattering of the perturbation across the sequence and the many missing signals certainly suggest that the bound state is not identifiable in the NMR spectra (no signal at the 1:2 ratio in Figure 5g). Changing the salt concentration or temperature may change the exchange regime.

Figure 5g shows the changes in peak intensity of the $^{13}\text{C}^{15}\text{N}$ -labelled $\text{Ca}^{2+}/\text{CaM}$ when titrated with unlabelled PYK2 KFL. We have quantified the affinity between $\text{Ca}^{2+}/\text{CaM}$ and KFL with ITC and MST to be $\sim 1 \mu\text{M}$ (Figure 2). Given the concentration of the labelled protein of $100 \mu\text{M}$, we have 94 % of molecules in the dimeric state in the sample. Therefore, the observed drop in peak intensity stems from the peak broadening as a result of the formation of the 2:2 complex. This plot is based on the lowest temperature we tried [10°C]. Increasing the temperature led to peak disappearance even before the 1:2 ratio. Given the limited solubility of the KFL we were also limited in the range of salt concentrations we could use. However, given the concentrations and affinities, we already expect to have 95 % of protein in a complex at the 1:1 ratio that we used to identify interacting residues. Hence the bound state is observable.

To respond to this comment, we have now more clearly indicated the concentrations of the proteins used in our NMR measurements:

“At the $250 \mu\text{M}$ concentration used for each of the proteins, and given the association K_d of $\sim 1 \mu\text{M}$, approximately 96 % of the labelled $\text{Ca}^{2+}/\text{CaM}$ was bound to PYK2 KFL₇₂₈₋₈₃₉.” (paragraph *PYK2-KFL binds to CaM in a process involving more than one linear peptide association*).

Concerning the CSP values, we agree to state more clearly the limitations of the data. Hence we have added in the Results section: “Although the mapping cannot be done with high confidence based on these data, considerably more hydrophobic CaM residues appear to be involved compared with the findings for titrations without Ca^{2+} .”

Additionally, we also amended the legend of the Figure 5i to better indicate the limitations of our analysis: “Blue residues are those assumed contributing only to dimerization (based on CSPs), and red residues are those that only contribute to CaM binding (in absence or presence of Ca^{2+}). Prolines and unassigned residues are marked in bold black. Note that the confidence of the residue mapping is low because of the fuzzy nature of the binding events.

2. Similar goes for the monomer-dimer equilibrium. There are not any zooms on peaks that move during the dilution so it is not possible to assess the data. Please include this to show saturation.

We agree that such a figure is useful. We have therefore prepared this figure (Supplementary Figure 8 in the revised manuscript), which shows the movements of selected CSPs, which we highlighted in zoomed-in figures.

Supplementary Figure 9: $[^1\text{H}, ^{15}\text{N}]$ HSQC overlay for $[^{13}\text{C}, ^{15}\text{N}]$ PYK2 KFL₇₂₈₋₈₃₉ at 100 μM (blue) and 10 μM (red) to analyse CSPs upon changes in the monomer:dimer ratio.

Here, again, intensity ratios and R2 values should be recorded so the correct dimerization interface is mapped. There are several data in the literature on disordered dimers (see e.g. Danielsson et al., *Biochemistry* 2008), so this is not a first observation made.

As noted above, we have now added the intensity ratios to the manuscript, and amended the Results, Discussion and Figure legends accordingly.

It is indeed not unprecedented that IDRs form homodimers and we agree that a more extended review of the current literature was required. We have done so in the *Discussion* part of the revised version: “Self-association of disordered regions has already been described. In many cases, however, homomultimerization is associated with stable long-range contacts [54], transitions to folding (e.g. α -synuclein) [55], MazE [56], papillomavirus E7 [57], electrostatic interactions between oppositely charged regions (e.g. HMGA2) [58], and/or defined residue-residue interactions (e.g. c-Myc PEST) [59], PQBP-1 K192Sfs*7 [60]. In our case, we demonstrated that the PYK2 KFL728-839 dimerizes without these features, in a fuzzy and flexible interaction that requires the full ~ 110 residue sequence. These features are reminiscent of those of the cytoplasmic domain of the T-cell receptor ζ chain or the chaperon 7B2. However, both may show a more promiscuous concentration-dependent oligomerization into dimers, tetramers and higher-order oligomers and are likely to adopt a more structured state upon binding to a client protein [61, 62]. Thus, the KFL expands the structural landscape of IDR self-association and may help analysing other IDRs for which homomultimers have been reported, but not fully structurally resolved (e.g. FEZ1) [63], and the N-terminal domain of Usp [64].”

3. Figure 1 provides the foundation for the study, however the Western Blot analyses are unclear, especially with respect to the quantification performed in Figure 1C. The representative image does not appear to match the quantified data, i.e., comparing the KFL700-841 band with PYK2 wt1-1009. It is also necessary to show the loading control for each blot. Western Blot analyses could be described better in the methods with respect to normalization to control bands.

Representative loading controls for the CaM sepharose bead experiments in **Fig. 1c** are shown in **Fig. 1b**. Because the expression levels of GFP-PYK2 constructs in HEK293 cells were variable depending on the deletion. It can be seen in Fig. 1b that deletion of a domain resulted in slightly higher expression than wild

type full-length Pyk2, while very large deletions including kinase, FERM and FAT domains which corresponded to the small KFL fragment were much more highly expressed, almost as much as GFP alone. Therefore to take into account these differences in the amount of GFP-PYK2 construct in the input when analyzing the amount bound to the beads, we divided the quantity in the beads (Fig. 1c, top panel) by the quantity in the input (Fig. 1b). Therefore the graph in Fig. 1c bottom panel does not correspond visually to the intensities in the blot of the top panel, but reflects more accurately the fraction of the GFP-PYK2 constructs that was bound to the beads.

We have now modified and completed the corresponding paragraph in the Material and Methods section (end of **Pull-down assay and immunoblot**) as follows:

“For quantification of experiments of binding of GFP-PYK2 (WT or deleted forms) to CaM beads in the presence of Ca²⁺ or EGTA, the amount of GFP-immunoreactivity associated with the beads was divided by the amount of GFP-immunoreactivity in the input for correction of the variable amounts of fusion protein depending on the constructs. For the experiments in which phosphorylation of PYK2 in response to depolarization was studied in the absence or presence of Cz, pY402 immunoreactivity was divided by total PYK2 immunoreactivity. Input and bead (Ca²⁺ and EGTA) fractions from recombinant purified protein were submitted to electrophoresis, stained using InVision His-Tag In-Gel stain and observed using BioRad Gel dock imager at 260 nm.”

4. The authors describe the KFL as extremely/exceptionally disordered. What are the underlying data that support the extreme/the exceptional disorder? It is disordered with a rather folded helix populated to at least 50% (or more). Why is it not just disordered? I think this is a more appropriate term. Also, why is this unusually flexible? Again, it is likely just disordered. There are others vaguely formulated terms as “peculiar features” (p.12 – what are they? And why are they peculiar?); “CaM caressing” – what does this mean? There are reports available on disordered complexes with disordered regions binding to e.g. ERK2 and to virus proteins (Milles, Cell; Hendus-Altenburger, BMC biology). How is the complex reported here different to those?

Concerning the ‘extreme’ disorder, we were referring to the atypical absence of either induction of secondary structure upon binding and the absence of long-range interactions in the dimer. However, we agree that it is better to simply refer to the KFL as “disordered”. Therefore, to address this comment, we took out all wording referring to an “extreme/exceptionally” disordered KFL. E.g. the title is now “PYK2 senses calcium through a disordered dimerization and calmodulin-binding element”. The abstract now reads: “our findings describe a flexible protein module that expands the paradigms for CaM binding and self-association”, instead of “highly flexible protein module that provides a new paradigm”. And the Discussion part now states: “our results describe a flexible protein module that expands the structural and functional range for CaM binding and self-association”, instead of “highly flexible protein module that provides a new paradigm for CaM binding and self-association”.

Re. “CaM caressing”: we agree that this is not a particularly well-chosen term. We changed it now to simply “disordered CaM binding”.

Re. “peculiar features”: In the revised version, we also better explained the atypical degree of disorder based on a better literature review (please see our detailed reply to the next question).

5. What is the new paradigm in CaM binding and why is this unusual/non-conventional? This is not clear from the text and because the bound state is likely invisible, details are missing that makes the conclusions hard to reach. In recent years, several studies have highlighted a huge diversity in CaM binding distinct from the helix-binding motifs, and there are reports on the existence of several different states interconverting even with different stoichiometries, including also binding to the calcium free state (see e.g. Lee...Ghooose, Protein Science 2019; Nunomoura et al., BJ, 2014; Sjogaard-Frich, Prestel et al., eLife, 2020), and oligomerization

involving CaM (Barros et al., IJMS 2019; Li...Sacks, JBC 2017). None of these reports are cited. A broader view on CaM binding is already emerging in the literature and should be referred to in the text.

We agree that the existing literature was not sufficiently covered, and the novelty in the CaM binding was not well explained. Following this comment, we have extended our discussion of the results with respect to the existing literature, and we thank this reviewer for their suggestion of published literature. We have also changed the wording from “new paradigm” to “extended the paradigms”.

The Discussion part now contains the following additional sections:

“The preservation of disorder in the association of unstructured proteins with their ligands has been observed previously [49], and several ligands were described to form “fuzzy” complexes with CaM. However, almost all of these CaM-interacting proteins still contained a well-defined helical region and/or fixed anchoring positions that stably associated with at least one CaM lobe [50, 51], even though several binding modes may be possible, depending on the conditions [52]. A closest known match to our system may be the association of a twenty-residue segment of the myelin basic protein (MBP₁₄₅₋₁₆₅) with CaM where a stable binding pose has not been identified [53]. However, MBP₁₄₅₋₁₆₅ is much shorter than PYK2 KFL₇₂₈₋₈₃₉, monomeric, and binds with a more than tenfold lower affinity and more ambiguous stoichiometry to CaM. Therefore, the association of PYK2 KFL₇₂₈₋₈₃₉ with CaM may present an extreme case within the spectrum of fuzzy CaM complexes.”

and

CaM has already been shown to promote dimerization in other ligands. For example, one CaM molecule can tether two molecules of the estrogen receptor or of the Na⁺/H⁺ exchanger NHE1 by simultaneously binding one molecule with each, its N- and C-terminal lobe, forming a 2:1 complex [65, 66]. However, conversely to these previous examples, CaM reinforces the already strong PYK2 KFL₇₂₈₋₈₃₉ dimer in a fuzzy 2:2 association.

6. In line with these issues, the claimed fuzzy interaction has not been supported by data. Relaxation measurements, diffusion, relaxation dispersion or molecular simulations, or smFRET data is needed to conclude this. The issue is if this interaction is just dynamic (fast on-off rates) or if there are multiple conformations of the bound state, that all contribute to the binding (which then suggest the interaction to be fuzzy). Please support these claims with data. If no folding occurs on binding, how do you explain the favorable enthalpy change?

As already mentioned above, relaxation experiments work (and were previously performed) in systems where the K_d was 10-100 fold higher than in our system. In these systems, a significant monomeric population can be obtained within the protein concentration limits for low-noise NMR data recording. This is not the case for the KFL with a dissociation K_d of less than 1 μM. In addition to the supplementary NMR analyses now included in the manuscript (intensity plots, comparison of spectra recorded at 700 and 950 MHz) we also support our claims using additional orthogonal biophysical measurements: Initially, we already had provided SAXS and CD measurements showing that the dimeric KFL is flexible and only contains some helical regions. We have now added CD spectra that we recorded for the KFL region alone, CaM alone and KFL bound to CaM. These spectra further support that the KFL does not gain in secondary structure upon binding. We have also added DSF data to show that the KFL region does not display an unfolding transition within the temperature range measured (from 24 to 100 °C). Thus, whereas our NMR measurements may not be bullet-proof in providing our claims, we believe that the NMR measurements together with all the other analyses types provide a compelling body of data.

Re. The favourable enthalpy: In this section (page 6), we compared PYK2 binding to CaM in the absence of Ca²⁺ with binding in the presence of Ca²⁺. Hence, we compared two systems in their bound states. We wrote

“ITC showed that the affinity increase in the presence of Ca^{2+} resulted from a favourable enthalpy gain (i.e. ionic and charged interactions) despite a loss in entropy (i.e. degrees of freedom of the system).” This observation is in agreement with our model where the flexible KFL and CaM form additional contacts when Ca^{2+} is present. Although more of these additional contacts are hydrophobic, the Ca^{2+} /CaM association also has (in total) a higher number of polar/charged associations (~34) compared to the reaction in presence of EGTA (~24), which would contribute to an overall more favourable enthalpy (please see Supplementary Table 3). Concerning the unfavourable entropy loss, our CD spectra and NMR analysis confirm that this loss is not the result of increased structuring of the KFL when binding to Ca^{2+} /CaM. Rather, the relative loss of entropy may stem from loss of dynamics of CaM and from loss of some degrees of freedom of the KFL N-term. Although the ITC enthalpy/entropy changes fit into our analysis, based on this reviewer’s comment we concluded that it is easier to take out the statement on entropy/enthalpy, because the explanation is lengthy but does not add significantly to our understanding of the mechanism, and our revised version already includes significantly more text, experiments and figures.

7. CD analyses have been provided for KFL domains alone, but it would be beneficial to add CD analysis of the PYK2 KFL domain interacting with CaM to demonstrate that KFL in the interaction remains disordered. This is particularly important as the bound state is inaccessible by NMR.

As suggested by this reviewer, we have now recorded CD spectra for the KFL region alone, CaM alone and KFL bound to CaM, with and without calcium (added as Supplementary Figure 5a,b). The analysis shows that there is no gain in secondary structuring in the complex (with and without Ca^{2+}), as the amount of helical residues in the complex is not higher than the individual contributions of each partner (see Supplementary Table 1, bottom panel). Hence, KFL does not gain in secondary structure upon binding and remains disordered. This information is now added to the section *PYK2 KFL forms a disordered composite motif for CaM-binding*.

Supplementary Figure 5: **(a)** CD spectra of PYK2 KFL₇₂₈₋₈₃₉ and CaM in the presence (left panel), and in the absence of Ca^{2+} (right panel). **(b)** CD spectrum of FAK KFL₇₇₆₋₈₄₁.

Construct	Predicted helical content	Helical content CD spectra		
		α -helix	β -sheets	Random coils
PYK2 KFL ₇₂₈₋₈₃₉ – Ca ²⁺	41%	36%	2%	62%
PYK2 KFL ₇₂₈₋₈₃₉ – EGTA	41%	36%	1%	63%
CaM - Ca ²⁺	-	65%	2%	33%
CaM - EGTA	-	61%	3%	36%
PYK2 KFL ₇₂₈₋₈₃₉ : CaM - Ca ²⁺	-	46%	3%	51%
PYK2 KFL ₇₂₈₋₈₃₉ : CaM - EGTA	-	49%	2%	49%
FAK KFL ₇₇₆₋₈₄₁	35%	27%	2%	71%

Construct	No. of residues	No. of helical residues
PYK2 KFL ₇₂₈₋₈₃₉ – Ca ²⁺	128	46
CaM - Ca ²⁺	164	107
PYK2 KFL ₇₂₈₋₈₃₉ : CaM - Ca ²⁺	292	134
PYK2 KFL ₇₂₈₋₈₃₉ – EGTA	128	46
CaM - EGTA	164	100
PYK2 KFL ₇₂₈₋₈₃₉ : CaM - EGTA	292	143

Supplementary Table 1. CD analysis. Top: Secondary structure content predicted by sequence analysis PSIPRED[2], and derived from CD data using the CAPITO web-server (capito.uni-jena.de; [7]) with JASCO ASCII output. Bottom: Helical content analysis for PYK2 KFL_{728–839} , CaM and PYK2 KFL_{728–839} : CaM complex showing no gain in helicity upon CaM interaction

8. Figure 2B is highly oversaturated and does not show co-localization. Please fix and provide an overlay to make this clearer.

We have amended Figure 2B to avoid oversaturation, now also showing more clearly the co-localisation between different labels.

9. Is the dimer broken or intact in the complex? I think this is relevant to address as a dimer here suggest similar mechanism as seen e.g in CaM binding to the estrogen receptor and in other membrane proteins.

Two lines of evidence support that the dimer is not only intact, but that dimeric KFL binds synergistically to CaM: Our MST experiments show that the KFL dimerises more strongly in presence of Ca²⁺/CaM, and this observation is corroborated by FA (which is a good orthogonal method, as it derives the K_d by the size increase of the complex, whereas MST uses the change in thermophoresis based on changes of the hydration shell and other factors)(Figure 6b). To enhance the clarity of this statements, we have reworded the summarising sentence in the Discussion section to read: “Although the presence of Ca²⁺/CaM only increased PYK2 KFL₇₂₈₋₈₃₉ dimerization two to three-fold in our MST and FA experiments (Fig. 6b), this effect may be reinforced [...]”.

10. The dispute regarding the role of CaM in regulation of PYK2 is not explained well enough that the reader can understand and appreciate the potential clarification this work may contribute with, please include.

We have now further clarified the controversy in the introduction, and have added an explanatory subfigure to Supplementary Fig. 1a. :

“Independent investigations by two research groups led to the proposition of two different mechanisms: Kohno et al. proposed that Ca²⁺-loaded CaM (Ca²⁺/CaM) binds to a helix within the FERM domain, whereas Xie et al. suggested that Ca²⁺/CaM binds to a helical region of the kinase domain (Fig. 1a)[40, 41]. Each proposed site would require unfolding of either the FERM or the kinase domain, respectively, to be accessible for canonical CaM interactions (Supplementary Fig. 1a). “

Supplementary Figure 1. Proposed non-KFL-specific CaM interactions. (a) Top: Schematic overview of FAK and PYK2, showing their domain structure and sequence identity. The localization of the previously proposed Ca^{2+} /CaM binding motifs is indicated in yellow. Bottom, left: Position of the previously proposed Ca^{2+} /CaM binding motifs (coloured in yellow) within the three-dimensional structures of the FERM domain (green, as proposed by Kohno et al, BJ, 2008) and kinase domain (blue; as proposed by Xie et al. Mol. Endo, 2008). Bottom, right: In these locations, the proposed Ca^{2+} /CaM binding motifs would not be able to associate with CaM in a canonical way, where calcium (red)-bound CaM (gray) wraps around the helical ligand (yellow).

11. Does KFL bind calcium?

We have now tested a direct interaction between KFL and calcium using ITC, and found that there was no such interaction (**Supplementary Fig 5e** of the revised manuscript). We added the following statement: “However, PYK2 KFL₇₂₈₋₈₃₉ did not bind to Ca^{2+} in absence of CaM (Supplementary Fig. 5e).” in the section *PYK2 KFL forms a disordered composite motif for CaM-binding*.

Supplementary Figure 5e: ITC experiment probing the interaction of PYK2 KFL₇₂₈₋₈₃₉ with calcium.

12. Is the binding of KFL to calcium-free CaM biologically relevant? No binding to CaM is seen in the presence of EGTA in the full-length protein.

This is an interesting question. At this stage we lack experimental data to address it. However, “we speculate that a weak association occurring between Ca^{2+} -free CaM and the PYK2 KFL₇₂₈₋₈₃₉ could enrich CaM close to PYK2 prior to Ca^{2+} influx, and/or weaken the inhibitory association of the KFL with the FERM-kinase fragment. Thus, the Ca-free association could help prime the system for a faster response upon Ca^{2+} influx.” These two sentences are now included in the legend of Fig. 6.

13. The lack of binding of the FERM domain could be due to the fact that the recombinant protein is not correctly folded. What data can be included to show this? CD, SEC.

Following this comment, we have reproduced recombinant PYK2 and FAK domains and have used DSF and CD to investigate their folded state. DSF reveals a T_m of $\sim 58^\circ\text{C}$ for FAK and $\sim 43^\circ\text{C}$ for FERM. At 25°C , CD shows the spectrum expected for a folded protein with mostly alpha and some beta secondary structure. At 60°C , CD shows that PYK2 is mostly unfolded, and that FAK FERM retains some residual structuring, as expected from their T_m .

Accordingly, we have added the sentence: “Circular dichroism (CD) and differential scanning fluorimetry (DSF) confirmed that the recombinant PYK2 and FAK FERM domains were correctly folded under the conditions used (Supplementary Fig. 2a,b,c).” in the first section of the Results part.

Supplementary Figure 2: Circular dichroism (CD) spectra for FAK FERM₃₁₋₄₀₅ and PYK2 FERM₃₅₋₃₉₇ at (a) 25°C and (b) 60°C . (c) Differential scanning fluorimetry (DSF) curve for FAK FERM₃₁₋₄₀₅ (melting temperature, $T_m = 57.8^\circ\text{C}$) and PYK2 FERM₃₅₋₃₉₇ ($T_m = 43.4^\circ\text{C}$).

14. For all binding experiments, the number of replicates (n) should be stated in the figure legend and the errors listed explained (are they errors of the fit, propagated errors, or standard errors of the mean, etc)

We have added this information now in the figure legends of all binding experiments.

15. It is possible – and interesting - that there are flanking region effect as context is emerging as playing key roles in affinity for IDPs. This should be substantiated once the binding regions have been more firmly established.

Flanking regions (and their entropic contributions) are particularly important for associations that pin down the ligand onto the protein surface. In our case, the KFL remains dynamic/fuzzy when bound to CaM. Nonetheless, a systematic analysis of the changes in K_d upon adding or removing increasing amounts of flanking sequences might address whether flanking regions are still important in such a fuzzy interaction. However, in our case we have difficulties identifying soluble fragments of this construct. Moreover, a further extension of the KFL upstream or downstream would reach the proline-rich and other interaction motifs and hence complicate our analysis beyond the scope of this manuscript.

Minors.

p. 2: several linear interaction motifs – Do you mean short linear motifs?

Yes, that's what we meant. To clarify we have now extended the sentence to read: "several short linear interaction motifs".

p. 7. Primary sequence -> primary structure or just sequence

We have changed it to "sequence".

Figure 4: used 250 mM

The legend has been corrected to 250 μ M.

Where did the author purchase their peptides?

The peptides were purchased from GenScript. We have now mentioned this in the methods section.

No DTT (or TCEP) was added to the CD analyses, why not?

We have used 0.5 mM TCEP in all CD measurements and it has been rectified in the materials and methods.

Figures could be easier to understand: e.g. providing a color legend on figure 5F, 5H; better demarcation of calcium in figure 6D (left).

We have remade the Figure 5 including the color legend on Figures 5 b,d,g,h, as pasted below.

Concerning the better indication of calcium in figure 6d (left), this seems to be a misunderstanding, as there is no calcium in the left panel. To avoid this misunderstanding, we have now better indicated that the right panel of figure 6d results from the addition of calcium in the figure (through a change of the way in which Ca²⁺ is introduced) and in the figure legend [Left: In the absence of Ca²⁺].

Reviewer #3 (Remarks to the Author):

This review is for the submitted manuscript "PYK2 senses calcium through a highly disordered dimerization and calmodulin- binding element" written by Dr. Stefan Arold, et al. This manuscript is a well written, scientifically important study and I enthusiastically support publication.

The major finding of the paper is the identification of a novel CaM binding element Specifically, "PYK2 KFL is highly disordered and engages CaM through an ensemble of transient binding events." The authors also

found that calcium increases this association by promoting structural changes in CaM.

These claims are supported by strong experimental evidence including a series of high quality NMR experiments.

We thank this reviewer for their strong support for our manuscript

One potential addition that could strengthen the manuscript would be to add NMR-based dynamics analysis, as changes in dynamics is suggested in the text but not supported experimentally by NMR.

Thank you for this comment. We refer to dynamics explicitly only once in the revised manuscript, namely when we state that “These additional interactions [occurring in presence of calcium] enhance the binding affinity and markedly constrain the dynamics of CaM, but not of PYK2 KFL₇₂₈₋₈₃₉.” This statement results from our NMR-based observation that backbone amide resonances of CaM, but not of PYK2 KFL, were broadening rapidly when titrated with the ligand protein. To clarify this connection, we have added “as judged from line broadening” to this sentence.

In the revised version, we have now also added more experimental evidence to support that PYK2 KFL remains unfolded and flexible: We have added CD spectra showing that no additional secondary structure is formed upon binding of KFL to Ca²⁺/CaM, and DSF analysis to show that there is no unfolding transition for the PYK2 KFL. Moreover, we now show NMR spectra recorded at different frequencies (700 and 950 MHz) to rule out that the bound state is invisible due to intermediate exchange. Finally, we have also added the protein concentrations used, showing more clearly that the bound state is predominant (~94%) under the conditions used for NMR.

We also considered determining the transverse relaxation rate. This rate is known to be sensitive to the tumbling time, and hence to the molecular weight/shape of the solute species. However, the strength of the association is so high ($K_d=0.8 \mu\text{M}$) that even if we dilute the KFL in the sample to $10 \mu\text{M}$ (we cannot use less due to the sensitivity loss), the sample will still contain less than 20% of the monomeric state contributing to the R2. However, changing the protein concentration 5 times will significantly decrease the viscosity, likely masking the transverse relaxation rate differences between monomeric and dimeric mixture (approximately 94 % dimer at $100 \mu\text{M}$ and 82 % of the dimer at $10 \mu\text{M}$). Moreover, the error of the measurement will increase with lower protein concentration. We agree that others have used this type of measurement successfully, but for a system with a much higher dimerisation K_d (e.g. 100 fold higher in Danielsson et al., Biochemistry 2008). Therefore, although this type of measurement is in principle very powerful, it will have only very limited value for testing our conclusion. However, the existence of the dimer is further verified by many other measurements, such as MST, fluorescence anisotropy, AUC, SEC-MALS and SAXS (Figures 2,3 and Supp Figures 2,5,6,7).

Regardless, it is my belief that this work would be of interest to the field and provides an interesting perspective regarding the interaction of CaM with target proteins.

many thanks!

Reviewers' comments:

Reviewer #1 (Remarks to the Author):

The authors have addressed all my points, and I am satisfied with the responses and new experiments addressing the points of other reviewers.

Reviewer #2 (Remarks to the Author):

In their revision, Momin et al have made an additional series of experiment that substantiate their work and which for a large part have addressed the concerns raised by this reviewer. However, a few issues remain and there are a few comments related to the presentation of the new data.

1) While we agree that several different lines of evidence establish the existence the dimer, it was not the reason for the additional experiment suggested in 1), but rather to establish and support the dynamics of the complex. Please include the intensity ratios of the two spectra recorded at 700 MHz and 900 MHz in Figure 4, this will be rather informative. For Figure 4B bottom – the grey area around 760 does not have significant CSPs and should likely be removed. Instead there are CSP above IQR around 783 and 798 – it seems not to be consistent as to which region is included and which is not.

2) The CD data are important additions. I have two comments pertaining to this. First, CD data is typically recorded at 1-10 micromolar concentrations (hard to assess in the M&M, please change from mg/ml to M), and with Kds of 1 and 13 uM, the concentration of the bound state is likely low. Please decrease the pathlength to increase the concentration of the complex and include these data. Second, once done, to fully appreciate the data, please include a comparison (overlay) of the sum of the individual spectra of CaM and KFL to the spectrum recorded of the complex. If no persistent contacts exist, these spectra should be identical and superimposable. It seems so, but should be directly accessible and at appropriate saturation.

3) For Ca²⁺ binding to KFL, please modify the statement “..did not bind to Ca²⁺ with measurable affinity by ITC”. It may still bind weakly if addressed by e.g. NMR.

4) Thanks for including the zooms. I find it odd that they all move in the same direction and with the same CSP. Is there an explanation to this? It seems to indicate a mismatch in temperature or other issues. Please comment on this in the manuscript and also show in the zoom a peak that does not move.

5) Thank you for clarification on the Western blots. I think it would still be helpful to include the full blots as part of the supplemental figures.

6) Speculation about the binding to CaM would do better in the main text than in a figure legend (Figure 6). Please change

Reviewer #3 (Remarks to the Author):

The revised manuscript "PYK2 senses calcium through a disordered dimerization and calmodulin-binding element" submitted by corresponding author Arold et al., is in my opinion an excellent manuscript, ready for publication. The original manuscript as strong, detailing a novel CaM-binding event. The authors responded to and addressed the concerns of all reviewers in great detail. I have no reservations to recommending that this manuscript be published as is.

Detailed response to reviewers

Reviewer #1 (Remarks to the Author):

The authors have addressed all my points, and I am satisfied with the responses and new experiments addressing the points of other reviewers.

We would like to thank you for your time and expertise in reviewing our manuscript.

Reviewer #2 (Remarks to the Author):

In their revision, Momin et al have made an additional series of experiment that substantiate their work and which for a large part have addressed the concerns raised by this reviewer. However, a few issues remain and there are a few comments related to the presentation of the new data.

Thank you for your time and expertise in reviewing our manuscript, we have now addressed these comments in the revised version as outlined below:

1) While we agree that several different lines of evidence establish the existence the dimer, it was not the reason for the additional experiment suggested in 1), but rather to establish and support the dynamics of the complex. Please include the intensity ratios of the two spectra recorded at 700 MHz and 900 MHz in Figure 4, this will be rather informative. For Figure 4B bottom – the grey area around 760 does not have significant CSPs and should likely be removed. Instead there are CSP above IQR around 783 and 798 – it seems not to be consistent as to which region is included and which is not.

We have addressed both parts of the comment. First, we have calculated the intensity ratios of both spectra, measured at 700 MHz and at 950 MHz. We found no change in the intensity ratios in any particular regions across the polypeptide. Hence, as suggested, we have included the intensity ratio in Figure 4d (pasted below). Secondly, as suggested we have removed the gray area near residue 760, and have included additional gray areas near residues 783 and 798.

The updated Figure 4b,c,d is below:

Figure 4. (b) (Top) CSP occurring as a result of diluting $[^{13}\text{C}, ^{15}\text{N}]$ PYK2 KFL₇₂₈₋₈₃₉ from 100 μM (8 scans) to 10 μM (128 scans), monitored by 2D $[^1\text{H}-^{15}\text{N}]$ HSQC. The orange horizontal line indicates the median threshold for minor shifts and the horizontal green line and shows the interquartile range (IQR) threshold for major shifts. **(Bottom)** Plot of 2D $[^1\text{H}-^{15}\text{N}]$ HSQC intensity ratios of $[^{13}\text{C}, ^{15}\text{N}]$ PYK2 KFL₇₂₈₋₈₃₉. The ratios were calculated as 10 μM (128 scans) divided by 100 μM (8 scans). The intensity value at 100 μM was taken as a reference and used to normalise the intensity value at 10 μM to compensate for an overall loss of intensity upon dilution. Red dotted line indicates the median. Grey shaded zones indicate residue regions where significant CSPs correspond to regions of lower relative intensity **(c)** CSPs from **(b)** were mapped on a representative 3D structure of PYK2 KFL₇₂₈₋₈₃₉. Major shifts are marked in magenta, minor shifts are coloured in pink and prolines, overlapping and unassigned residues are marked in black. N and C indicate the N-terminal and C-terminal of the molecule. **(d)** Plot of 2D $[^1\text{H}-^{15}\text{N}]$ HSQC intensity ratios of $[^1\text{H}, ^{15}\text{N}]$ PYK2 KFL₇₂₈₋₈₃₉. The ratios were calculated as 100 μM (8 scans) measured on 700 MHz divided by 100 μM (8 scans) measured on 950 MHz.

2) The CD data are important additions. I have two comments pertaining to this. First, CD data is typically recorded at 1-10 micromolar concentrations (hard to assess in the M&M, please change from mg/ml to M), and with Kds of 1 and 13 μM , the concentration of the bound state is likely low. Please decrease the pathlength to increase the concentration of the complex and include these data. Second, once done, to fully appreciate the data, please include a comparison (overlay) of the sum of the individual spectra of CaM and KFL to the spectrum recorded of the complex. If no persistent contacts exist, these spectra should be identical and superimposable. It seems so, but should be directly accessible and at appropriate saturation.

Thank you for pointing this out. We have measured the CD spectra on smaller proteins at higher concentrations as suggested, ranging from 20 μM to 40 μM with 0.1 mm path-length cell. We have also updated all the concentrations changing them from mg/ml to M and updated the methods section with clearly mentioning concentrations for each protein. It is to be noted, that the PYK2 KFL₇₂₈₋₈₃₉ : CaM complex sample was measured at 40 μM , which is more than double the binding Kd in the absence of

Ca²⁺ (13μM), hence almost 70 % of the proteins should be in the bound state, even for this weakest association.

To address the second part of the comment, we have included a comparison (overlay) of the sum of the individual spectra of PYK2 KFL₇₂₈₋₈₃₉ and CaM, to the spectrum of the complex. As the reviewer mentioned, we note that the sum of individual spectra of both proteins fits well on the complex spectrum. We have updated the Supplementary Figure 5, as pasted below:

Supplementary Figure 5. Biophysical analysis of PYK2 and FAK KFL. CD spectra of PYK2 KFL₇₂₈₋₈₃₉ and CaM in the presence (a), and in the absence (b) of Ca²⁺. (c) CD spectrum of FAK KFL₇₇₆₋₈₄₁. Overlapped CD spectra of PYK2 KFL₇₂₈₋₈₃₉ : CaM complex in the presence (d) and absence (e) of Ca²⁺, on the sum of the individual spectra of PYK2 KFL₇₂₈₋₈₃₉ and CaM, as shown in (a) and (b).

3) For Ca²⁺ binding to KFL, please modify the statement “..did not bind to Ca²⁺ with measurable affinity by ITC”. It may still bind weakly if addressed by e.g. NMR.

We have now modified the statement in the revised manuscript which now reads as “PYK2 KFL₇₂₈₋₈₃₉ did not bind to Ca²⁺ in the absence of CaM with measurable affinity by ITC”.

4) Thanks for including the zooms. I find it odd that they all move in the same direction and with the same CSP. Is there an explanation to this? It seems to indicate a mismatch in temperature or other issues. Please comment on this in the manuscript and also show in the zoom a peak that does not move.

Thank you for pointing this out, it is indeed important to show peaks that do not move. We have updated the Supplementary Figure 9, which now includes the zoomed-in images of the peaks which do not move upon dilution (K801 and W821). In addition, we have also included more zoomed-in peaks (L808 and V833). As shown in the picture, not all peaks move in the same directions, and we do have other shift variations (L808) including the peaks which do not move at all (K801, W821). Hence, these peak shifts are not related to temperature changes in the measurement. The updated Supplementary Figure 9 is pasted below.

Supplementary Figure 9. [¹H,¹⁵N] HSQC overlay for [¹³C,¹⁵N] PYK2 KFL₇₂₈₋₈₃₉ at 100 μM (blue) and 10 μM (red) to analyse CSPs upon changes in the monomer:dimer ratio. To support that CSPs were not simply a general result of small changes in temperature or pH, we also displayed two peaks that did not move upon dilution (K801 and W821).

5) Thank you for clarification on the Western blots. I think it would still be helpful to include the full blots as part of the supplemental figures.

We have now distinctly presented the Western Blots in the main figures, separating a Figure 1c which were two different gels next to each other in the previous version of the manuscript. Also, as suggested by the reviewer, we have now included all the uncropped immunoblots in Supplementary Figure 1b,c,d.

We have marked the cropped areas for readers to understand the exact boundaries on the immunoblots cropped and presented in main Fig 1b,c.
The updated figure 1b,c is pasted below:

Figure 1: Identification of the CaM binding site in PYK2. (b) Representative immunoblot of input fractions of the indicated GFP-tagged PYK2 constructs immunolabelled with GFP and tubulin antibodies. **(c) Top:** Representative GFP immunoblot of GFP-PYK2 constructs associated with the CaM Sepharose beads labelled as in **(b)**. **Bottom:** Graphical representation of GFP immunoblot densitometry associated with beads normalized by corresponding input, presented as fold change using PYK2 WT Ca²⁺ bead fractions as a reference. Bars correspond to the mean of 4-7 independent experiments, ±SEM.

The uncropped immunoblots as shown in Fig 1b,c above are included in the manuscript as Supplementary Figure 1b,c as pasted below:

Supplementary Figure 1: Uncropped images of the representative immunoblot of input fractions of the indicated GFP-tagged PYK2 constructs immunolabeled with **(b)** GFP and **(c)** tubulin antibodies as shown in Fig. 1b. **(d)** Uncropped images of representative GFP immunoblot of GFP-PYK2 constructs associated with the CaM Sepharose beads as shown in Fig 1c. Boxed areas represent the cropped part of the immunoblots.

6) Speculation about the binding to CaM would do better in the main text than in a figure legend (Figure 6). Please change

We have now moved our proposed model into the Discussion part, and have amended the legend of Figure 6 accordingly.

Reviewer #3 (Remarks to the Author):

The revised manuscript "PYK2 senses calcium through a disordered dimerization and calmodulin-binding element" submitted by corresponding author Arold et al., is in my opinion an excellent manuscript, ready for publication. The original manuscript is strong, detailing a novel CaM-binding event. The authors responded to and addressed the concerns of all reviewers in great detail. I have no reservations to recommending that this manuscript be published as is.

We would like to thank you for your time and expertise in reviewing our manuscript.

REVIEWERS' COMMENTS:

Reviewer #2 (Remarks to the Author):

The authors has satisfactorily addressed my concerns.

Detailed response to reviewers

Reviewer #2 (Remarks to the Author):

The authors has satisfactorily addressed my concerns

We thank you for your time and expertise in reviewing our manuscript.